# Epigenetic targeting of bromodomain protein BRD4 counteracts cancer cachexia and prolongs survival

Marco Segatto [1], Raffaella Fittipaldi[1], Fabrizio Pin[2], Roberta Sartori[3,4], Kyung Dae Ko[5], Hossein Zare[5], Claudio Fenizia[1], Gianpietro Zanchettin[6], Elisa Sefora Pierobon[6], Shinji Hatakeyama [7], Cosimo Sperti[6], Stefano Merigliano[6], Marco Sandri[4,5], Panagis Filippakopoulos[8,9], Paola Costelli[2], Vittorio Sartorelli[5] & Giuseppina Caretti [1]

Cancer cachexia is a devastating metabolic syndrome characterized by systemic inflammation and massive muscle and adipose tissue wasting. Although it is responsible for approximately one-third of cancer deaths, no effective therapies are available and the underlying mechanisms have not been fully elucidated. We previously identified the bromodomain and extra-terminal domain (BET) protein BRD4 as an epigenetic regulator of muscle mass. Here we show that the pan-BET inhibitor (+)-JQ1 protects tumor-bearing mice from body weight loss and muscle and adipose tissue wasting. Remarkably, in C26-tumor-bearing mice (+)-JQ1 administration dramatically prolongs survival, without directly affecting tumor growth. By ChIP-seq and ChIP analyses, we unveil that BET proteins directly promote the muscle atrophy program during cachexia. In addition, BET proteins are required to coordinate an IL6-dependent AMPK nuclear signaling pathway converging on FoxO3 transcription factor. Overall, these findings indicate that BET proteins may represent a promising therapeutic target in the management of cancer cachexia.

[1] Department of Biosciences, Universita' degli Studi di Milano, Via Celoria 26, 20133 Milan, Italy. [2] Department of Clinical and Biological Sciences, Unit of General and Clinical Pathology, University of Turin, 10124 Torino, Italy. [3] Department of Biomedical Sciences, University of Padova, 35131 Padova, Italy. [4] Venetian Institute of Molecular Medicine, 35131 Padova, Italy. [5] Laboratory of Muscle Stem Cells and Gene Regulation, NIH/NIAMS, 50 South Drive, Bethesda, MD, USA. [6] Department of Surgery, Oncology and Gastroenterology, 3rd Surgical Clinic, University of Padua, 35122 Padova, Italy. [7] Musculoskeletal Disease Area, Novartis Institutes for BioMedical Research Basel, Novartis Pharma AG, 4056 Basel, Switzerland. [8] Structural Genomics Consortium, Old Road Campus Research Building, Nuffield Department of Medicine, University of Oxford, Oxford OX3 7DQ, UK. [9] Ludwig Institute for Cancer Research, Old Road Campus Research Building, Nuffield Department of Medicine, University of Oxford, Oxford OX3 7DQ, UK. Correspondence and requests for materials should be addressed to G.C. (email: giuseppina.caretti@unimi.it)

Cancer cachexia is a multifactorial metabolic syndrome characterized by systemic inflammation and muscle and adipose tissue wasting, which lead to weight loss despite adequate nutritional support[1].

It accompanies majority of cancer patients with advanced disease, and has a stronger prevalence at earlier cancer stages in solid tumors, such as gastric, pancreatic, lung, colorectal, and head and neck[2]. Cachexia is predictive of reduced response to tumor therapies and is associated with increased risk of complications that correlate with fatal outcome[1,2].

Though cachexia accounts for nearly 30% deaths of cancer patients[3,4], no effective treatment is currently available. It was recently reported that pharmacological intervention aimed at blocking muscle wasting effectively prolongs survival in experimental models of cancer cachexia[5,6]. In humans, therapeutic intervention with a single agent targeting exclusively systemic inflammation, low food intake, or catabolic pathways regulating muscle mass has proven ineffective to counteract cachexia[1,7]. A combinatorial approach is advocated for successful management of cachexia, in order to concurrently address different facets of the syndrome, such as inflammation and muscle atrophy[8]. IL6, the ubiquitin proteasome system, and autophagy have been individually investigated for potential therapeutic intervention, with limited success[9–12]. Thus, the precise elucidation of molecular pathways underlying cancer cachexia is crucial for the identification of novel key targets, which regulate multiple pathophysiological aspects of the disease and that become appealing for therapeutic intervention. Transcription factors and epigenetic regulators are of particular interest in this scenario, as they hold the potential to reprogram multiple transcription programs in different tissues and orchestrate coordinated transcription in different districts[1,7,12].

The onset of cachexia is thought to depend both on tumor-secreted factors and on the host response. In skeletal muscle and adipose tissues, cytokines and other factors (i.e., tumor necrosis factor alpha (TNFα), interleukin 6 (IL6), interleukin 1 beta (IL1β), myostatin, activin, parathyroid hormone-related protein (PTHrP)) derived by the tumor, the host immune system, or mesenchymal tissues trigger an intracellular signaling cascade, which translates into transcriptional changes in the gene-expression programs, eliciting catabolic responses[1]. In cachexia-induced muscle atrophy, two main catabolic pathways are activated: the ubiquitin-proteasome degradation system, through transcriptional upregulation of MAFbx/Atrogin-1 and MuRF1 ubiquitin-ligases, and autophagy[13–15]. The transcription factor FoxO3 orchestrates the expression of key factors of both catabolic pathways, in several muscle wasting conditions[16]. Although co-activators and chromatin regulators likely take part in the transcriptional modulation of catabolic genes, the epigenetic mechanisms underlying their activation in cachexia are poorly elucidated.

We recently reported that, in an in vitro model of glucocorticoid-induced skeletal muscle atrophy, the bromodomain protein BRD4 plays a role in controlling myotubes size. Accordingly, BRD4 blockade with the bromodomain and extra-terminal domain (BET) proteins inhibitor (+)-JQ1 prevents atrophy of dexamethasone-treated myotubes[17]. Taking into account these findings and the compelling number of reports highlighting the ability of BET inhibitors to challenge progression of several forms of cancer[18–21] and the promising prospect of developing (+)-JQ1 derivatives in the clinic[22], we asked whether BET proteins are involved in the transcriptional activation of catabolic genes in cancer cachexia, and whether the BET small inhibitor (+)-JQ1 may have a beneficial effect by preventing muscle wasting.

We report that BRD4 directly associates with regulatory regions of key catabolic genes in skeletal muscle and that BRD4 recruitment dramatically increases during muscle wasting. BRD4 blockade by (+)-JQ1 administration results in BRD4 loss from catabolic genes loci and muscle sparing.

In addition, BETs blockade reduces systemic IL6 and PTHrP levels, and adipose tissue loss.

Importantly, BRD2 also participates to the regulation of a subset of catabolic genes in skeletal muscle and to the transcriptional modulation of IL6 and PTHrP. IL6 reduction prevents AMPK activation in muscles of cachectic mice. Notably, we describe that during cancer cachexia AMPK directly regulates transcription of catabolic genes, by occupying their promoter regions. Additionally, accumulation of the active form of AMPK in the nuclear compartment of cachectic myofibers contributes to FoxO3 phosphorylation, which promotes transcriptional activation. Overall, BET blockade by (+)-JQ1 administration orchestrates a dual control on the expression of genes involved in muscle atrophy, by impairing BRD4 and BRD2 direct occupancy at catabolic genes and by restraining the IL-6/AMPK/FoxO3 axis activation. Thus, BET blockade in skeletal muscle and tumor ameliorate skeletal muscle integrity in C26-tumor-bearing mice.

## Results

**C26 adenocarcinoma cells growth is resistant to JQ1.** To explore the involvement of the BET inhibitor (+)-JQ1 in the activation of catabolic genes and muscle wasting in cancer cachexia, we first established (+)-JQ1 sensitivity in nine cell lines commonly employed in experimental models of cancer cachexia, by assessment of (+)-JQ1 impact on viability after 72 h of (+)-JQ1 treatment, at three different doses (0.1, 0.5, and 1 μM). Eight of the nine cell lines displayed sensitivity to (+)-JQ1 for viability, while C26 colon adenocarcinoma cells were only mildly affected by the treatment, at the three concentrations (Fig. 1a). In agreement with (+)-JQ1's well-described antineoplastic function, we observed a significant decrease ($p < 0.0001$, unpaired $t$-test) in tumor weight when mice were inoculated with melanoma B16 tumors and treated with (+)-JQ1 (20 mg/kg/day) for 10 days before sacrifice (Fig. 1b). Because of C26 colon carcinoma cells resistance to (+)-JQ1 treatment in vitro, C26 xenografts were selected as a suitable model to in vivo investigate the effect of (+)-JQ1 administration on cancer cachexia. To challenge BET protein function, tumor-bearing mice were treated with the BET inhibitor (+)-JQ1 or its inactive enantiomer (−)-JQ1 at two doses (20 and 50 mg/kg) daily, starting the day following tumor cells inoculation, for 12 days. Notably, tumor weight was not affected by (+)-JQ1 administration in this experimental model (Fig. 1c). In addition, (+)-JQ1 administration did not impact c-Myc protein levels[21], Caspase-3 activation[23] (Fig. 1d) phosphorylated histone H3[24] (Supplementary Fig. 1), suggesting that C26 adenocarcinoma cells proliferation and apoptosis were not influenced by (+)-JQ1 administration.

**JQ1 prevents muscle loss and prolongs survival in cachexia.** Tumor-bearing animals treated with either the vehicle or the inactive enantiomer (−)-JQ1 (20 and 50 mg/kg/day) lost body weight starting 10 days after C26 cells inoculation, and reached 22% of total body weight at day 12. In contrast, body weight was comparable in (+)-JQ1-treated mice and control animals throughout the treatment and at the day sacrifice, suggesting that (+)-JQ1 administration prevented wasting, at both doses (20 and 50 mg/kg/day) (Fig. 2a). Likewise, starting from day 9, food intake was dramatically reduced in C26-bearing mice treated with vehicle or (−)-JQ1, but not in the case of (+)-JQ1-treated tumor-bearing mice and control animals until the day of sacrifice

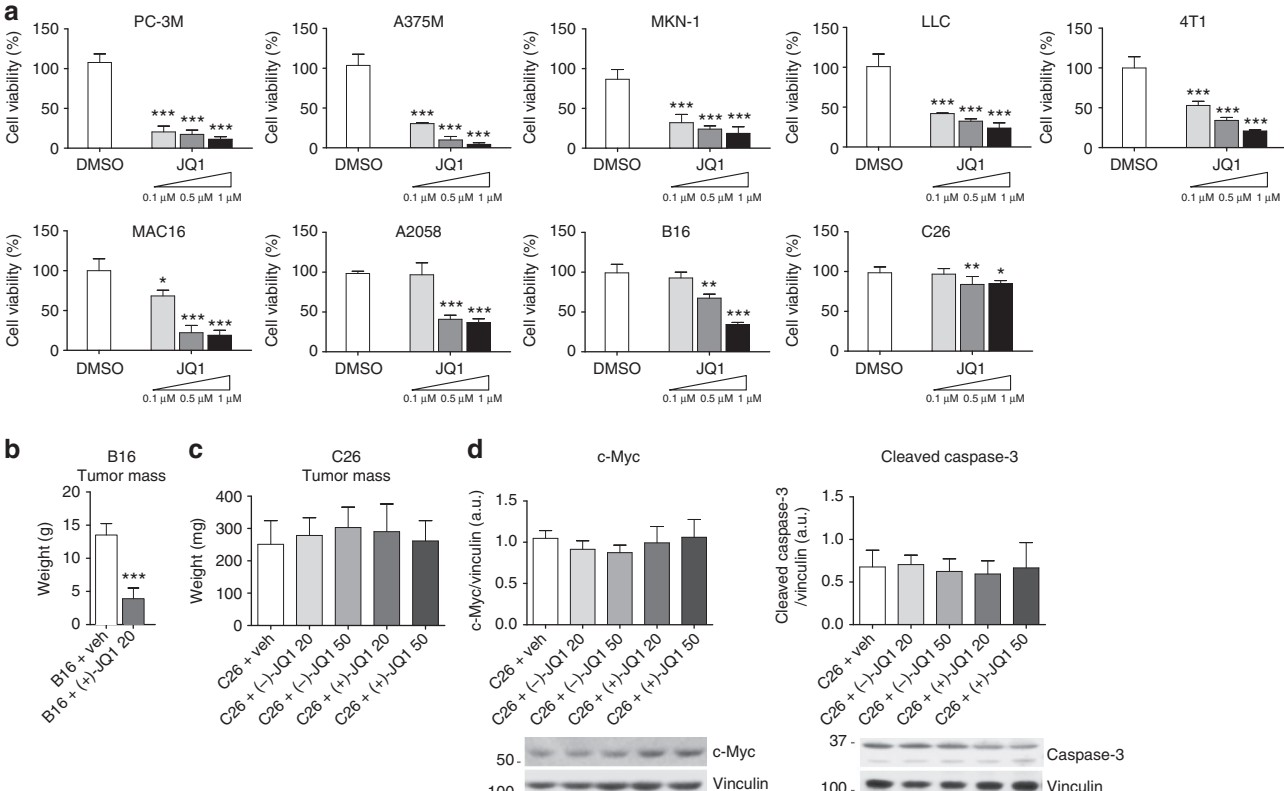

**Fig. 1** Response to JQ1 in cancer cell lines used to study cancer cachexia. **a** Viability rates of JQ1-treated cells, calculated by measuring the viable cell number after 72 h in culture. Results are normalized to the proliferation of vehicle/DMSO-treated cells, set to 100% ($n = 3$). Statistical analysis was performed by using one-way ANOVA followed by Tukey's post hoc test. Data represent means $\pm$ SD. $*p < 0.05$; $**p < 0.01$; $***p < 0.001$ vs. DMSO. **b** Ten days after B16-tumor cells inoculation, mice were treated either with vehicle or (+)-JQ1 (20 mg/kg/day) for 10 days. Tumor weight after necropsy is shown. Animals per group: $n = 7$. Data represent means $\pm$ SD. Statistical analysis was performed by using Student's unpaired $t$-test. $***p < 0.001$ indicates statistical significance vs. B16 + veh. **c** Tumor weight at the day of sacrifice (day 12) in C26-tumor-bearing mice (animals per group: $n = 10$). Data represent means $\pm$ SD. Statistical analysis was performed by using one-way ANOVA followed by Tukey's post hoc test. **d** c-Myc and cleaved-Caspase-3 levels were quantified by immunoblots using tumor whole extracts (animals per group: $n = 4$)

(Fig. 2b). Importantly, even at the lowest dose (20 mg/kg/day), (+)-JQ1 treatment was associated with a significantly ($p = 0.0004$, log-rank (Mantel–Cox) test) prolonged survival compared to vehicle-treated animals, with a shift of the median survival from 16 to 28 days (Fig. 2c). To provide a more therapeutically relevant preclinical model, we started (+)-JQ1 administration at a moderate cachexia stage, when C26-tumor-bearing mice displayed an average 10% of weight loss. Surprisingly, (+)-JQ1 treatment was able to significantly prolong survival ($p = 0.0005$, log-rank (Mantel–Cox) test) and partially reverse cachexia, shifting median survival from 16 to 28 days (Fig. 2c).

When JQ1 was administered the day after tumor cell inoculation, cachexia was delayed and C26-tumor-bearing mice started to lose weight at day 24 (Supplementary Fig. 2, gray line). Furthermore, when treatment was started during moderate cachexia, C26-tumor-bearing mice partially reversed weight loss few days after (+)-JQ1 treatment started, and their weight was stabilized till day 23, further delaying cachexia (Supplementary Fig. 2, blue line).

To further evaluate body sparing of (+)-JQ1-treated animals, we analyzed body composition. (+)-JQ1 treatment fully prevented tibialis anterior (TA), extensor digitorum longus (EDL), soleus, and epididymal fat mass loss observed in vehicle and (–)-JQ1-treated C26-tumor-bearing animals (Fig. 2d–g). Consistently with metabolic alterations observed in cancer cachexia[25,26], C26-tumor implantation determined a strong increase in plasma LDL cholesterol, which was

hindered by (+)-JQ1 administration (Supplementary Fig. 3b). No differences were detected in both total cholesterol and HDL cholesterol among the six experimental groups (Supplementary Fig. 3a, c). Other organs, such as heart were unaffected by (+)-JQ1 treatment (Supplementary Fig. 3d); testis was used as a positive control for (+)-JQ1 efficacy since testis mass was reported to be sensitive to (+)-JQ1 treatment[27] (Supplementary Fig. 3e). Morphological analysis of the TA confirmed that (+)-JQ1 administration counteracted the reduction in myofiber size observed in vehicle-treated C26-tumor-bearing mice (Fig. 2h; Supplementary Fig. 4a, b). One of the main hallmarks of cancer-associated cachexia, such as impairments in physical activity, was prevented upon (+)-JQ1 administration and (+)-JQ1-treated animals performed significantly ($p = 0.0003$, one-way ANOVA) better in the treadmill test, when compared to cachectic vehicle-treated animals (Fig. 2i). We conclude that (+)-JQ1 administration to C26-tumor-bearing mice prevents muscle and fat tissue loss and blocks cachexia progression. The outcome is a prolonged survival through a mechanism unrelated to tumor growth.

**JQ1 prevents catabolic pathways activation in muscle.** During cancer cachexia two major degradative pathways mediate skeletal muscle loss: the ubiquitin/proteasome system (UPS) and the autophagy pathway[13,15]. As expected, fast and slow myosin heavy chain proteins were dramatically reduced both in vehicle and (–)-JQ1-treated mice, 12 days after tumor implantation.

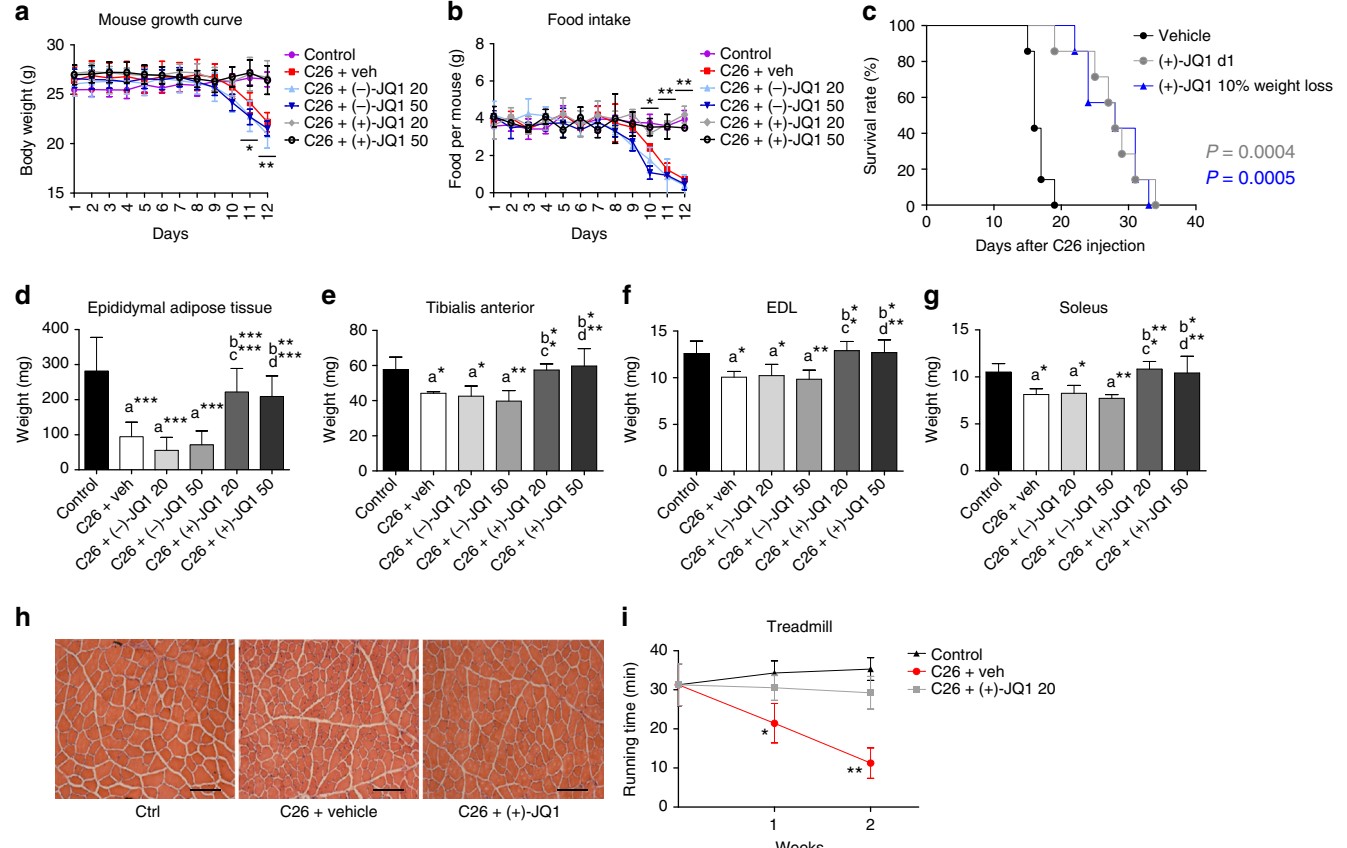

**Fig. 2** JQ1 prolongs survival, and prevents muscle and white fat wasting. **a** Body weight chart of control, vehicle, (−)-JQ1, and (+)-JQ1-treated C26-tumor-bearing mice. Day 0: tumor implantation. Day 1: JQ1 treatment started. Age-matched normal control mice were treated with vehicle (animals per group: n = 10). Data represent means ± SD. Statistical significance refers to (+)-JQ1-treated animals vs. vehicle and (−)-JQ1-treated animals. **b** Chart displaying food intake in the six animal groups described in **a**. Data represent means ± SD. Statistical significance refers to (+)-JQ1-treated animals vs. vehicle and (−)-JQ1-treated animals. **c** Survival rate of C26-tumor-bearing mice was measured in two animal groups: vehicle (veh) and (+)-JQ1 20 mg/kg/day. Treatment was started either at day 1 (gray line) or when the average weight loss was 10% (blue line) (animals per group: n = 7). **d** Epididymal adipose tissue was weighted at day of sacrifice (day 12) in the six animal groups (animals per group: n = 10). Data represent means ± SD. **e–g** TA, EDL, and soleus were weighted 12 days after C26 cell implantation (animals per group: n = 10). Data represent means ± SD. **h** TA muscle morphology of Ctrl and C26-tumor-bearing mice treated with vehicle and (+)-JQ1 (animals per group: n = 3). (+)-JQ1 was administered at the dose of 20 mg/kg/day. Scale bar: 100 μm. **i** Treadmill test was performed on control and C26-tumor-bearing mice treated with (−)-JQ1 and (+)-JQ1 (20 mg/kg/day) at days 7 and 14 (animals per group: n = 4). Data represent means ± SD. Statistical significance refers to (+)-JQ1-treated animals vs. vehicle and (−)-JQ1-treated animals. *p < 0.05; **p < 0.01; ***p < 0.001. "a" indicates statistical significance compared to control; "b" indicates statistical significance compared to C26 + vehicle; "c" indicates statistical significance compared to C26 + (−)-JQ1 20 mg/kg/day; "d" indicates statistical significance compared to C26 + (−)-JQ1 50 mg/kg/day. Statistical analysis was performed by using one-way ANOVA followed by Tukey's post hoc test in **a**, **b**, **d–g**, and **i** log-rank (Mantel–Cox) test in **c**

Conversely, TA from (+)-JQ1-treated animals displayed fast and slow myosin levels comparable to control animals (Fig. 3a; Supplementary Fig. 5a). Accordingly, in vitro degradation assays showed that fast myosin degradation was sustained in vehicle and (−)-JQ1-treated muscle extracts, but was abrogated in muscle extracts from (+)-JQ1-treated C26-tumor-bearing mice (Fig. 3b). (+)-JQ1 administration prevented transcriptional activation of ubiquitin-ligases responsible for skeletal muscle protein degradation, MuRF1 (p < 0.0001, one-way ANOVA), MAFbx/Atrogin1 (p < 0.0001, one-way ANOVA), and Fbxo30/Musa1 (p = 0.0001, one-way ANOVA), which were significantly increased in vehicle and (−)-JQ1-treated tumor-bearing mice compared to the control animal group (Fig. 3c, d; Supplementary Fig. 5b). MAFbx/Atrogin1 protein levels were maintained at basal levels in muscles from (+)-JQ1-treated C26-tumor-bearing mice, although they were significantly upregulated (p < 0.0001, one-way ANOVA) in vehicle and (−)-JQ1-treated cachectic mice (Fig. 3e). Autophagy undergoes both short- and long-term activation during cachexia[9,15,28]; accordingly autophagy-related

transcripts Bnip3, LC3b, GABARAPL1, Atg7, and CathepsinL were considerably increased in vehicle and (−)-JQ1-treated C26-tumor-bearing mice. (+)-JQ1 treatment hindered the transcriptional upregulation of autophagy-related genes (Fig. 3f–h; Supplementary Fig. 5c, d), and hampered the conversion of LC3 from the unlipidated (LC3-I) to the lipidated form (LC3-II) (Fig. 3i). AMPK-dependent activatory phosphorylations of Ulk1 and Beclin1, which are involved in autophagy initiation, were increased in vehicle and (−)-JQ1-treated tumor-bearing mice, but not in the (+)-JQ1-treated animals (Fig. 3j; Supplementary Fig. 5e). Taken together, these results reveal that (+)-JQ1 treatment prevents UPS and autophagy induction in TA muscles of C26-tumor-bearing mice.

**JQ1 prevents BETs occupancy at catabolic genes in cachexia.** To understand the role of BRD4 in transcriptional regulation of tumor-induced skeletal muscle wasting, we mapped the genome-wide distribution of BRD4 in skeletal muscles from

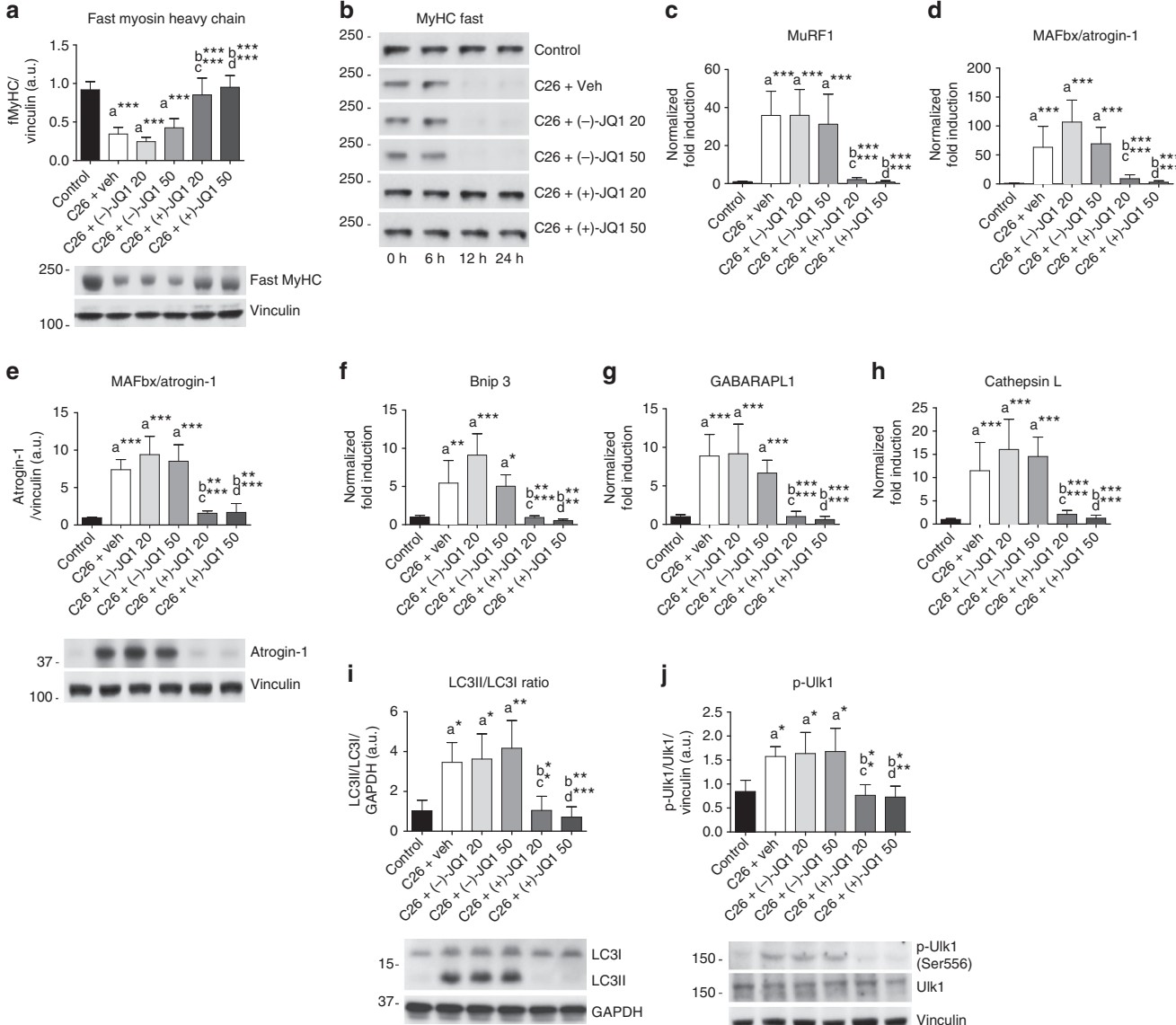

**Fig. 3** Muscle proteolysis and autophagy are hindered by JQ1 administration. **a** Representative immunoblot showing fast myosin heavy chain (MyHC) protein expression in TA whole extracts from the six animal groups. Upper panel shows quantification of immunoblot bands from four animals per group. Data represent means ± SD. **b** TA extracts from control and C26-tumor-bearing mice (animals per group: $n = 3$) were used in an in vitro degradation assay to reveal fast MyHC degradation in control vs. tumor-bearing mice treated with vehicle or JQ1 (−/+). **c, d** Total RNA was extracted from TA muscles from control and C26-tumor-bearing mice (animals per group: $n = 10$) treated with vehicle or JQ1 (−/+) and expression levels of MuRF1, MAFbx/Atrogin-1 were measured by quantitative RT-PCR. Data represent means ± SD. **e** Representative western blot of MAFbx/Atrogin-1 on TA whole extracts (animals per group: $n = 4$). Bands quantifications are shown in the upper panel. Data represent means ± SD. **f–h** Quantitative RT-PCR of autophagy genes (Bnip3, GABARAPL1, Cathepsin L) from control and C26-tumor-bearing mice. Ten animals were used for each experimental group. Data represent means ± SD. **i, j** Representative western blot for LC3 and p-Ulk1 in TA extracts of control and C26-tumor-bearing mice (animals per group: $n = 4$). Upper panel: quantification of normalized band intensity. Data represent means ± SD. Statistical analysis was performed by using one-way ANOVA followed by Tukey's post hoc test. $*p < 0.05$; $**p < 0.01$; $***p < 0.001$. "a" indicates statistical significance compared to control; "b" indicates statistical significance compared to C26 + vehicle; "c" indicates statistical significance compared to C26 + (−)-JQ1 20 mg/kg/day; "d" indicates statistical significance compared to C26 + (−)-JQ1 50 mg/kg/day

control and C26-tumor-bearing mice treated with either (−)-JQ1 (20 mg/kg/day) or (+)-JQ1 (20 mg/kg/day) by ChIP-seq. ChIP-seq data were analyzed using MACS peak calling tool[29] at the FDR level of 5% ($q$-value = 0.05). Genes were assigned using a proximity distance of 50 kb from gene body. Correlation analysis of BRD4 peaks in cachexia/control and cachexia/cachexia (+)-JQ1 treatment revealed that many regions gaining BRD4 during cachexia have reduced BRD4 upon (+)-JQ1 treatment (Fig. 4a). In control skeletal muscle, BRD4 peaks were assigned to 11,581

genes and localized in intergenic (37%), intronic regions (35%), 5′/3′UTR (20%), and exons (7%). BRD4 enrichment was evident in 492 assigned genes in skeletal muscle from C26-tumor-bearing mice. Acquired BRD4 peaks lay primarily in intronic regions (59%), suggesting that they may represent regulatory regions (Fig. 4b). Gene Ontology analysis of biological processes on genes displaying increased BRD4 recruitment in cachexia revealed enrichment in categories related to inflammatory pathways and catabolic processes, key events occurring in skeletal muscle during

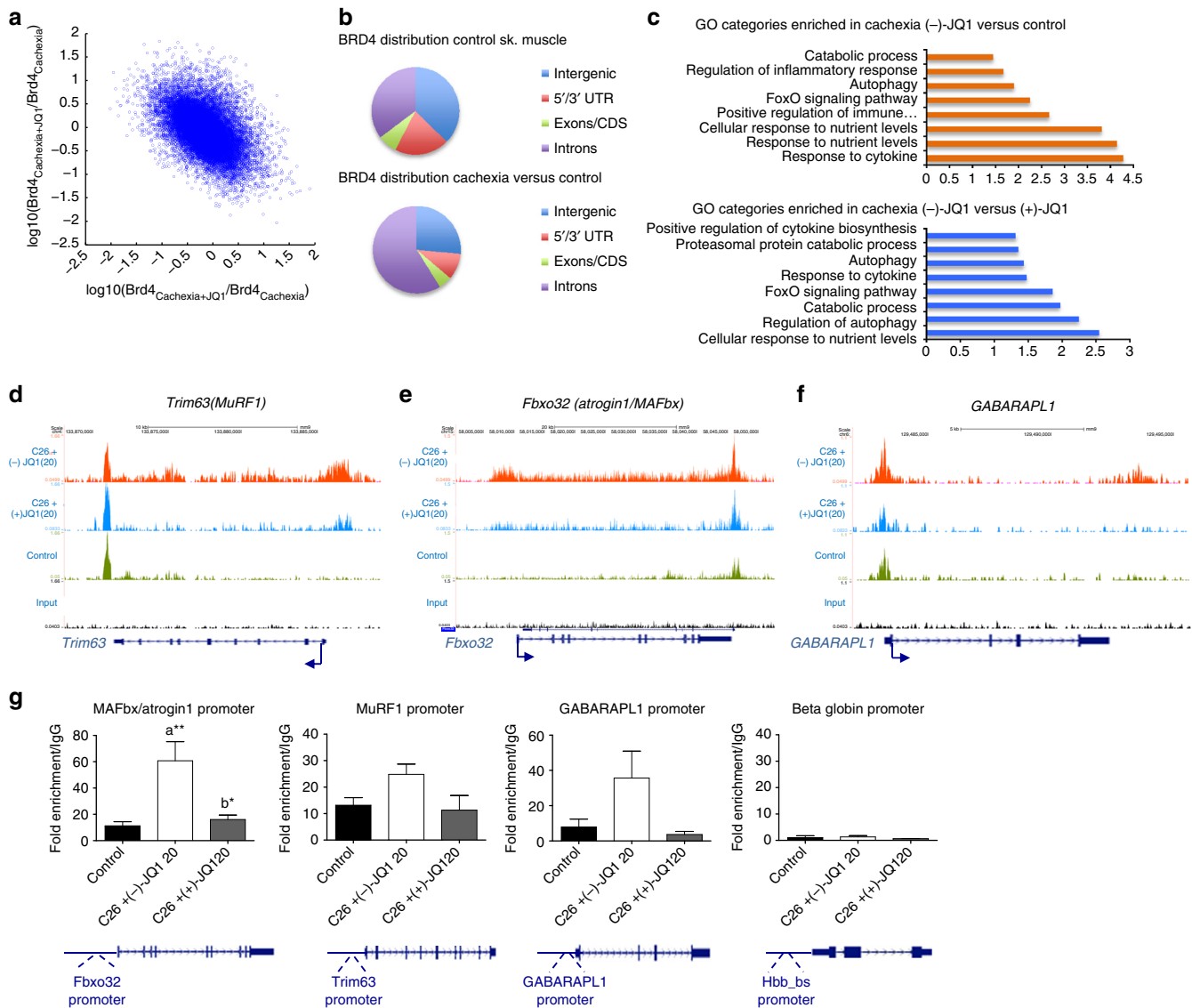

**Fig. 4** BRD4 occupies catabolic genes regulatory regions in cachexia and its recruitment is impaired by JQ1 administration. **a** Scatter plot of log10 fold changes between $BRD4_{cachexia}$ and $BRD4_{ctrl}$ vs. fold changes between $BRD4_{cachexia+JQ1}$ and $BRD4_{cachexia}$ for all enriched regions. The negative slope indicates the inhibitory effect of (+)-JQ1 on BRD4 occupancy reversing the cachexia effect on BRD4 occupancy. **b** Pie chart showing BRD4 distribution in control skeletal muscle (left panel) and cachectic muscles vs. control (right panel). **c** Selected functional categories of genes with higher BRD4 in skeletal muscle of (−)-JQ1-treated C26-tumor-bearing mice vs. control muscles (top panel) and in muscles from (−)-JQ1 vs. JQ1(+)-treated C26-tumor-bearing mice (bottom panel). **d**–**f** BRD4 ChIP-seq tracks of *Trim63 (MuRF1)*, *Fbxo32 (Atrogin1/MAFbx)*, and *GABARAPL1* loci. Bottom to top: input, control muscles, muscles from C26-tumor-bearing mice treated with (+)-JQ1 and (−)-JQ1 (top). **g** RNAPolII ChIP qPCR at promoters of muscle catabolic genes. *Beta globin* gene was used as a negative control. IgG was used as a reference. $n = 3$. Statistical analysis was performed by using one-way ANOVA followed by Tukey's post hoc test. Data represent means ± SEM. *$p < 0.05$; **$p < 0.01$. "a" indicates statistical significance compared to control; "b" indicates statistical significance compared to C26 + (−)-JQ1 20 mg/kg/day

cancer cachexia (Fig. 4c; Supplementary Data 1). BRD4 occupancy was higher in 136 assigned genes in skeletal muscles from cachectic mice, when compared to muscles from (+)-JQ1-treated animals. Notably, these genes were enriched in several categories related to catabolism and inflammation, suggesting that JQ1 treatment reduced BRD4 engagement in pro-cachectic genes (Fig. 4c; Supplementary Data 1).

In atrophying muscles, BRD4 was increased at known regulatory regions and in the gene bodies of key atrophy-related genes (*Fbxo32* and *Trim63*) encoding for the muscle-specific E3 ubiquitin ligases MAFbx/Atrogin1 and MuRF1. BRD4 recruitment at the E3 ubiquitin-ligases MuRF1 and MAFbx/Arogin1 genes was reduced in skeletal muscles of (+)-JQ1-treated

C26-tumor-bearing mice (Fig. 4d, e). Likewise, BRD4 occupancy at the *GABARAPL1* gene increased in muscles from cachectic mice and was reduced upon (+)-JQ1-treatment (Fig. 4f). To explore the molecular mechanisms underlying *MAFbx/Atrogin1*, *MuRF1*, and *GABARAPL1* transcriptional regulation, we investigated RNA Polymerase II (RNAPolII) recruitment at these loci. RNAPolII association at the promoters of *MAFbx/Atrogin1*, *MuRF1*, and *GABARAPL1* increased in muscles from (−)-JQ1-treated C26-tumor-bearing mice when compared to control muscles, in agreement with the enhanced RNA transcript levels (Fig. 3c, d, g). Conversely, RNAPolII occupancy at these promoters was comparable in skeletal muscles from (+)-JQ1-treated C26-tumor-bearing mice and control muscles, suggesting

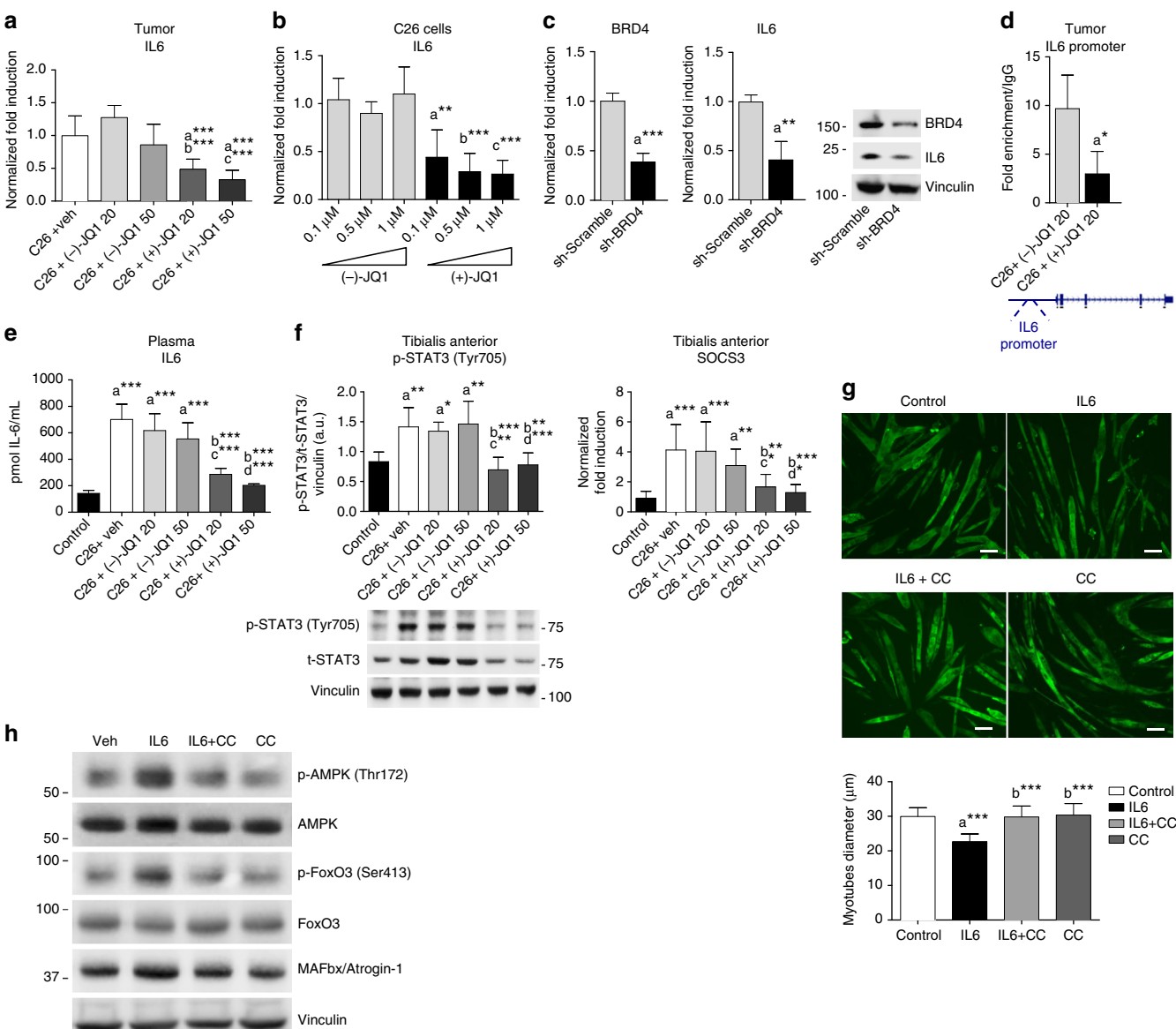

**Fig. 5** JQ1 administration lowers circulating and tumor-producing IL6 levels. **a** Tumor-derived IL6 transcripts (means ± SD) were measured by quantitative RT-PCR. Statistical analysis: one-way ANOVA, Tukey's post hoc test. ***$p < 0.001$. "a", "b", "c" indicate statistical significance vs. C26 + vehicle, C26 + (−)-JQ1 20 mg/kg/day and C26 + (−)-JQ1 50 mg/kg/day, respectively. **b** C26 cells were treated with different doses of (−)-JQ1 and (+)-JQ1, and IL6 transcripts were measured by qRT-PCR. Mean ± SD, $n = 5$. Statistical analysis: one-way ANOVA, Tukey's post hoc test. **$p < 0.01$; ***$p < 0.001$. "a", "b", "c" indicate statistical significance vs. C26 cells treated with (−)-JQ1 0.1, 0.5, and 1 μM, respectively. **c** C26 cells were transduced with Sh-scramble or Sh-BRD4 retroviruses, and BRD4 and IL6 transcripts were measured by qRT-PCR. Mean ± SD, $n = 5$. Statistical analysis: Student's unpaired $t$-test. **$p < 0.01$; ***$p < 0.001$. Right panel: BRD4 and IL6 immunoblots of Sh-Scramble and Sh-BRD4 C26 cells. **d** BRD4 recruitment at IL6 promoter was measured by ChIP in tumors from (−)-JQ1- and (+)-JQ1-treated (20 mg/kg/day) mice. Means ± SEM, $n = 3$. Statistical analysis: Student's unpaired $t$-test. *$p < 0.05$. **e** ELISA on plasma IL6 in control and C26-bearing mice treated with vehicle, (−)-JQ1 and (+)-JQ1 at 20 and 50 mg/kg/day. $n = 5$. Statistical analysis: one-way ANOVA, Tukey's post hoc test. ***$p < 0.001$. Data represent means ± SD. "a", "b", "c", and "d" indicate statistical significance vs. control, C26 + vehicle, C26 + (−)-JQ1 20 mg/kg/day and C26 + (−)-JQ1 50 mg/kg/day, respectively. **f** p-STAT3 and SOCS3 were analyzed in TAs. Left panel: p-STAT3 (Tyr705) and t-STAT3 were evaluated by western blot. Means ± SD, $n = 6$. Right panel: SOCS3 levels were quantified by quantitative RT-PCR. Means ± SD, $n = 8$. Statistical analysis: one-way ANOVA, Tukey's post hoc test. *$p < 0.05$; **$p < 0.01$; ***$p < 0.001$. "a", "b", "c", and "d" indicate statistical significance as in **e**. **g** C2C12 myotubes were treated with IL6, Compound C/IL6, or Compound C alone for 48 h. Cells were immunostained with MyHC (MF20) antibody. Scale bar: 50 μm. Lower panel: average diameter size of myotubes. Means ± SD, $n = 3$. Statistical analysis: one-way ANOVA, Tukey's post hoc test. ***$p < 0.001$. "a", "b" indicates statistical significance vs. control and IL6-treated cells, respectively. **h** Cells were treated as in **g** and p-AMPK (Thr172), total AMPK, p-FoxO3 (Ser413), total FoxO3, and Atrogin1/MAFbx were analyzed by western blot. GAPDH serves as loading control ($n = 3$).

that BRD4 blockade prevented RNAPolII association at these promoters (Fig. 4g). Taken together, these findings suggest that BRD4 directly takes part in the transcriptional regulation of *MAFbx/Atrogin1*, *MuRF1*, and *GABARAPL1* genes, and that its engagement correlates with RNAPolII recruitment at regulatory

regions of atrogenes and autophagy genes. Because of BRD2 overlapping regulation in several subsets of BRD4 targets[30,31], we investigated BRD2 ability to engage *MAFbx/Atrogin1*, *MuRF1*, and *GABARAPL1* loci, during cachexia. BRD2 associates with *MAFbx/Atrogin1* and *GABARAPL1* promoters during cachexia,

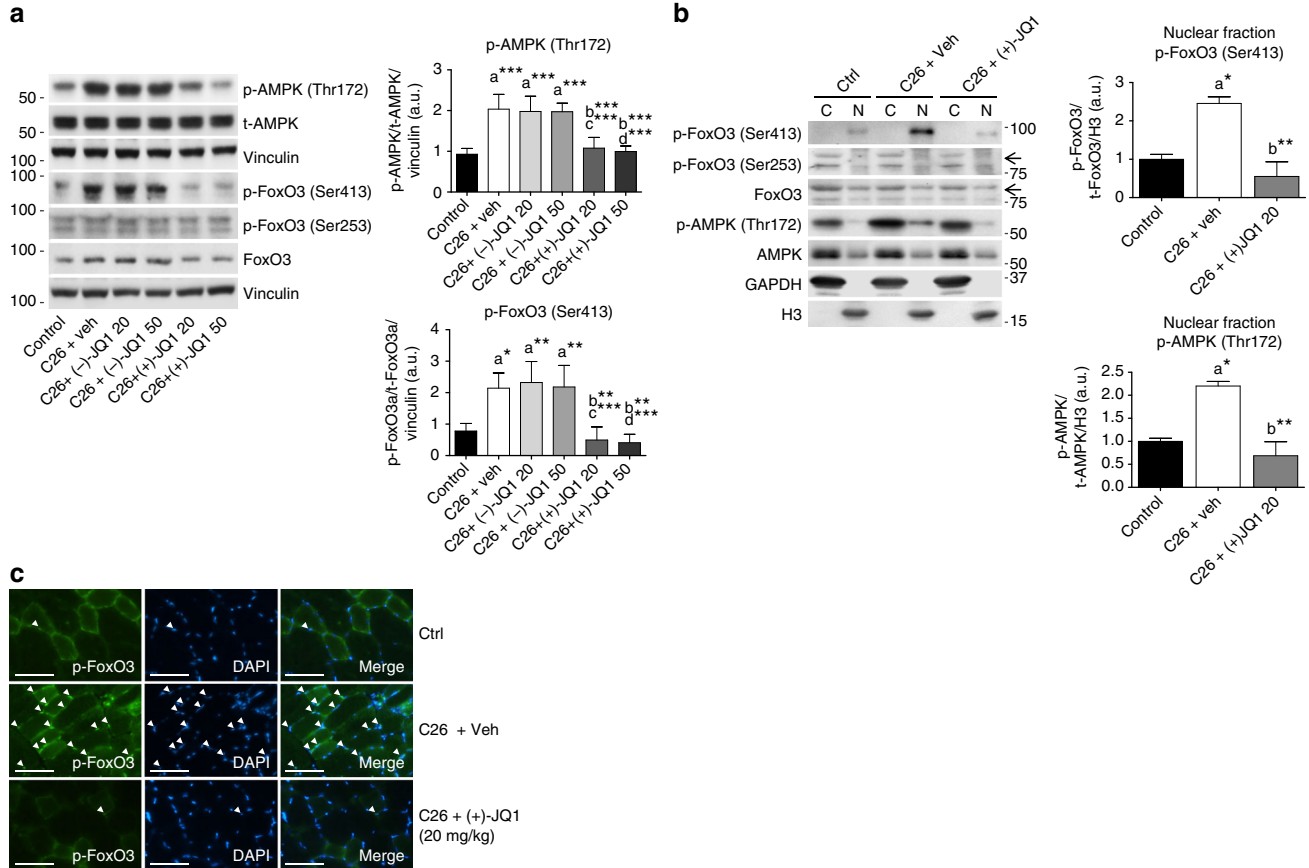

**Fig. 6** p-FoxO3Ser413 and p-AMPKThr172 levels increase in muscles from cachectic mice. **a** Left panel: representative western blot of p-AMPK (Thr172) and total AMPK, p-FoxO3 (Ser413), p-FoxO3 (Ser253), total FoxO3 levels in TA extracts from control animals and tumor-bearing mice treated with vehicle, (−)-JQ1, and (+)-JQ1. Vinculin serves as loading control. Middle panel: average normalized values of band intensity for p-AMPK (Thr172). Right panel: average normalized values of band intensity for p-FoxO3 (Ser413) (animals per group: n = 4–6). Data represent means ± SD. Statistical analysis was performed by using one-way ANOVA followed by Tukey's post hoc test. *p < 0.05; **p < 0.01; ***p < 0.001. "a" indicates statistical significance compared to control; "b" indicates statistical significance compared to C26 + vehicle; "c" indicates statistical significance compared to C26 + (−)-JQ1 20 mg/kg/day; "d" indicates statistical significance compared to C26 + (−)-JQ1 50 mg/kg/day. **b** Left panel: representative western blot analysis of p-FoxO3 (Ser413), p-FoxO3 (Ser253), total FoxO3, p-AMPK (Thr172) and total AMPK in nuclear/cytoplasmic extracts of TAs from control and tumor-bearing mice treated with vehicle or (+)-JQ1 (20 mg/ml/day). Three animals were used for each experimental group. Middle panel: average normalized values of band intensity for p-FoxO3 (Ser413) in nuclear fraction. Right panel: average normalized values of band intensity for p-AMPK (Thr172) in nuclear extracts. Statistical analysis was performed by using one-way ANOVA followed by Tukey's post hoc test. *p < 0.05; **p < 0.01. "a" indicates statistical significance compared to control; "b" indicates statistical significance compared to C26 + vehicle. **c** Frozen TA sections (animals per group: n = 3) were stained with antibodies raised against p-FoxO3 (Ser413) antibodies. Arrowheads point at nuclei positive for p-FoxO3. Scale bar: 50 μm

and this association is abrogated by (+)-JQ1 treatment (Supplementary Fig. 6). However, BRD2 was not recruited at *MuRF1* promoter and enhancer regions (Supplementary Fig. 6), suggesting that BRD4 is the principal BET protein-regulating *MuRF1* transcription, during cachexia.

**JQ1 perturbs the expression of tumor pro-cachectic factors.** Since pro-inflammatory cytokines play a critical role in the onset of cancer cachexia, we measured mRNA levels of IL6, TNFα, IL1β, and PTHrP[32,33] in C26 tumors isolated from mice treated with vehicle, (−)-JQ1 and (+)-JQ1 (20 and 50 mg/kg/day). No significant differences in IL1β (p = 0.7705, one-way ANOVA) and TNFα (p = 0.1252), one-way ANOVA) transcript levels were detected (Supplementary Fig. 7a, b). TNFα-dependent NFκB -p65 (Ser 536) phosphorylation was also unaffected by (+)-JQ1 treatment in TA skeletal muscle (Supplementary Fig. 7c). In contrast, IL6 (p < 0.0001, one-way ANOVA) and PTHrP (p = 0.0002, one-way ANOVA) levels were significantly decreased in tumors from (+)-JQ1-treated mice, when compared with tumors

from vehicle- and (−)-JQ1-treated animals (Fig. 5a; Supplementary Fig. 7d).

To assess the function of BRD4 in IL6 transcriptional regulation in C26 cells, we initially employed an in vitro approach. JQ1 treatment as well as BRD4 knockdown resulted in a prompt reduction of IL6 levels in C26 cells (Fig. 5b, c). Chromatin immunoprecipitation assays in tumors showed that BRD4 is recruited at the *IL6* promoter and that (+)-JQ1 treatment impairs BRD4 occupancy, suggesting that (+)-JQ1 plays a primary role in hindering IL6 expression in C26 adenocarcinoma tumors (Fig. 5d). Accordingly, we found that (+)-JQ1 administration significantly (p < 0.0001, one-way ANOVA) reduced IL6 circulating levels, which were significantly (p < 0.0001, one-way ANOVA) elevated in tumor-bearing mice treated with either vehicle or (−)-JQ1, when compared to control mice (Fig. 5e).

IL6 induces signal transducer and activator of transcription 3 (STAT3) phosphorylation and STAT3-dependent activation of target genes[34]. In TAs of (+)-JQ1-treated tumor-bearing mice, STAT3 activatory phosphorylation (p < 0.0001, one-way ANOVA) and expression of the STAT3 target suppressor of

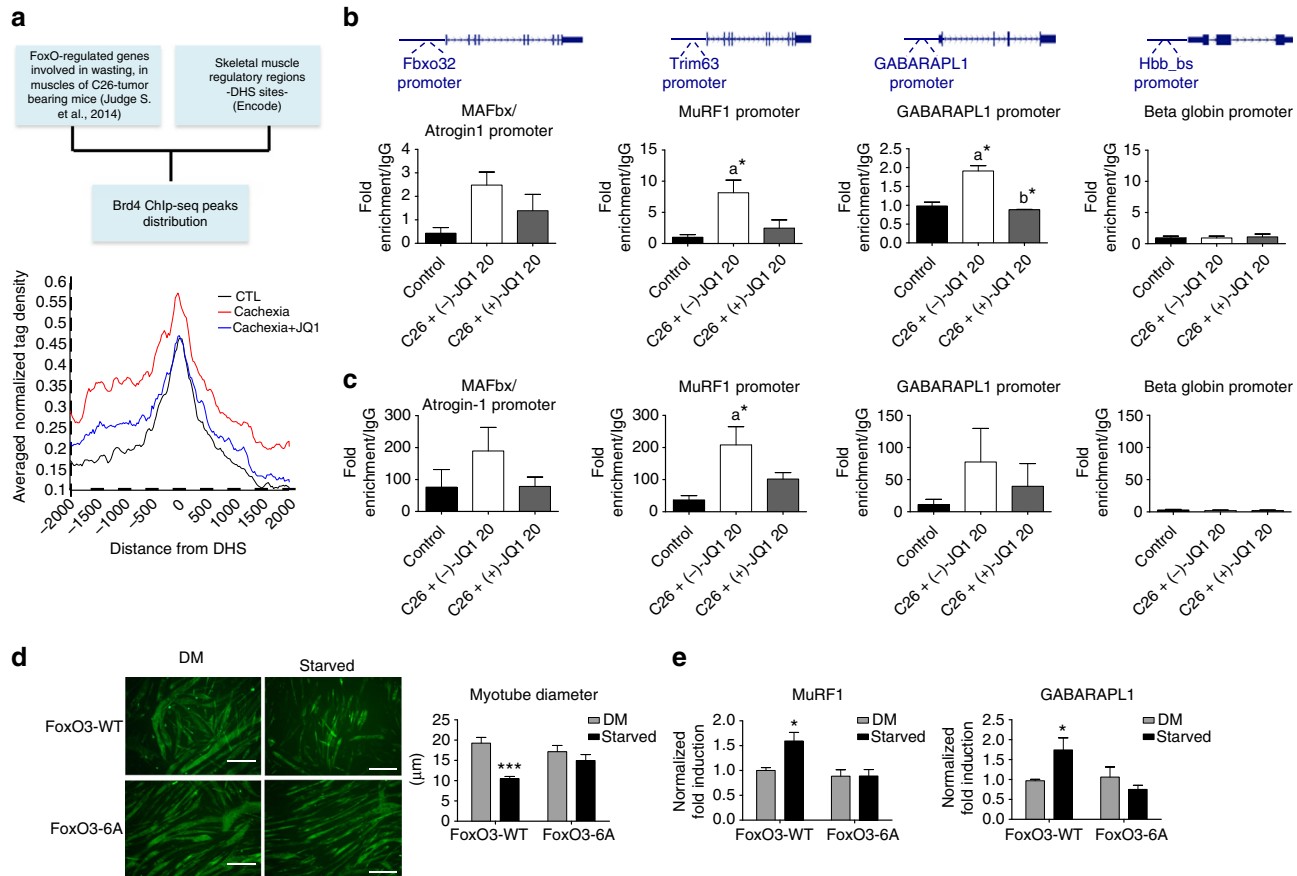

**Fig. 7** FoxO3 and AMPK directly associate to catabolic genes promoters in skeletal muscles of cachectic mice. **a** Left panel: scheme of the bioinformatics analysis used to generate the average profile of BRD4 ChIP-seq tag counts around ±2 kb of DHSs neighboring a subset of FoxO-regulated genes in control, Cachexia, and Cachexia/JQ1+ mice (right panel). **b** FoxO3 ChIP qPCR at promoters of muscle catabolic genes and *beta globin* gene as a negative control. IgG was used as a reference; $n = 3$. Data represent means ± SEM. Statistical analysis was performed by using one-way ANOVA followed by Tukey's post hoc test. *$p < 0.05$; "a" indicates statistical significance compared to control; "b" indicates statistical significance compared to C26 + (−)-JQ1 20 mg/kg/ day. **c** p-AMPK(Thr172) ChIP qPCR at promoters of muscle catabolic genes. *Beta globin* gene is employed as a negative control. IgG was used as a reference. Animals per group: $n = 3$. Data represent means ± SEM. Statistical analysis was performed by using one-way ANOVA followed by Tukey's post hoc test. *$p < 0.05$; "a" indicates statistical significance compared to control. **d** C2C12 were transduced with FoxO3-WT or FoxO3-6A retroviruses and allowed to differentiate for 4 days. Myotubes were then maintained in a DM medium with 25 mM glucose or starved in a medium with 0.5 mM glucose and 0.1% horse serum, for 24 h. Immunofluorescence was performed with ant-MyHC antibody (MF20). Scale bar: 50 μm. Right panel: mean diameter of FoxO3-WT and FoxO3-6A myotubes is shown. Data represent means ± SEM. Statistical analysis was performed by using one-way ANOVA followed by Tukey's post hoc test. ***$p < 0.001$ vs. FoxO3-WT in DM. **e** Total RNA from myotubes treated as in **d** was extracted, and MuRF1 and GABARAPL1 transcript levels were measured by quantitative RT-PCR. Data represent means ± SD. Statistical analysis was performed by using one-way ANOVA followed by Tukey's post hoc test. *$p < 0.05$ vs. FoxO3-WT in DM

cytokine signaling 3 (SOCS3) ($p < 0.0001$, one-way ANOVA) were significantly reduced (Fig. 5f). JQ1 treatment concurrently decreased PTHrP mRNA expression in C26 cells (Supplementary Fig. 7e). BRD4 occupied the *PTHrP* promoter in tumors from (−)-JQ1-treated animals, and this association was abrogated in tumors isolated from (+)-JQ1-treated animals (Supplementary Fig. 7f). Accordingly, plasma levels of PTHrP were also reduced upon (+)-JQ1 treatment (Supplementary Fig. 7g). Together with BRD4, BRD2 was implicated in modulation of the inflammatory response in bone marrow-derived macrophages and in sepsis[35,36] and in the regulation of metabolic pathways[37]. Thus, we investigated whether BRD2 contributed to the transcriptional regulation of pro-cachectic factors such as IL6 or PTHrP in the tumor. We have found that BRD2 is recruited at *IL6* and *PTHrP* promoters and (+)-JQ1 treatment reduces its engagement, although less effectively, when compared to BRD4 (Supplementary Fig. 8). Overall, these data support a role for BET proteins in modulating circulating IL6 and PTHrP levels and suggest that the

BET inhibitor (+)-JQ1 restrains cancer cachexia through the reduction of IL6 and other cachectic factors.

**IL6-mediated AMPK activation promotes muscle atrophy.** Activation of the 5′-adenosine monophosphate-activated protein kinase (AMPK) signaling pathway has recently been implicated in cancer cachexia[38–40]. AMPK plays a crucial role in energy homeostasis and is a sensor of the metabolic state during cancer cachexia[38]. Notably, AMPK phosphorylates FoxO3 at residues Thr179, Ser399, Ser413, Ser555, Ser588, and Ser626[41]. Increasing lines of evidence hint for a central role of IL6 in AMPK activation: IL6 knockout mice display decreased AMPK activation in skeletal muscle and adipose tissue[42], IL6 treatment leads to AMPK activation in C2C12 myotubes[38], and systemic administration of IL6 receptor antibody attenuates AMPK activation in Apc[Min/+] cachectic mice[43]. To in vitro recapitulate the impact of IL6 on muscle cells, we treated C2C12 myotubes with IL6 and observed myotube atrophy, as revealed by a reduction in myotube

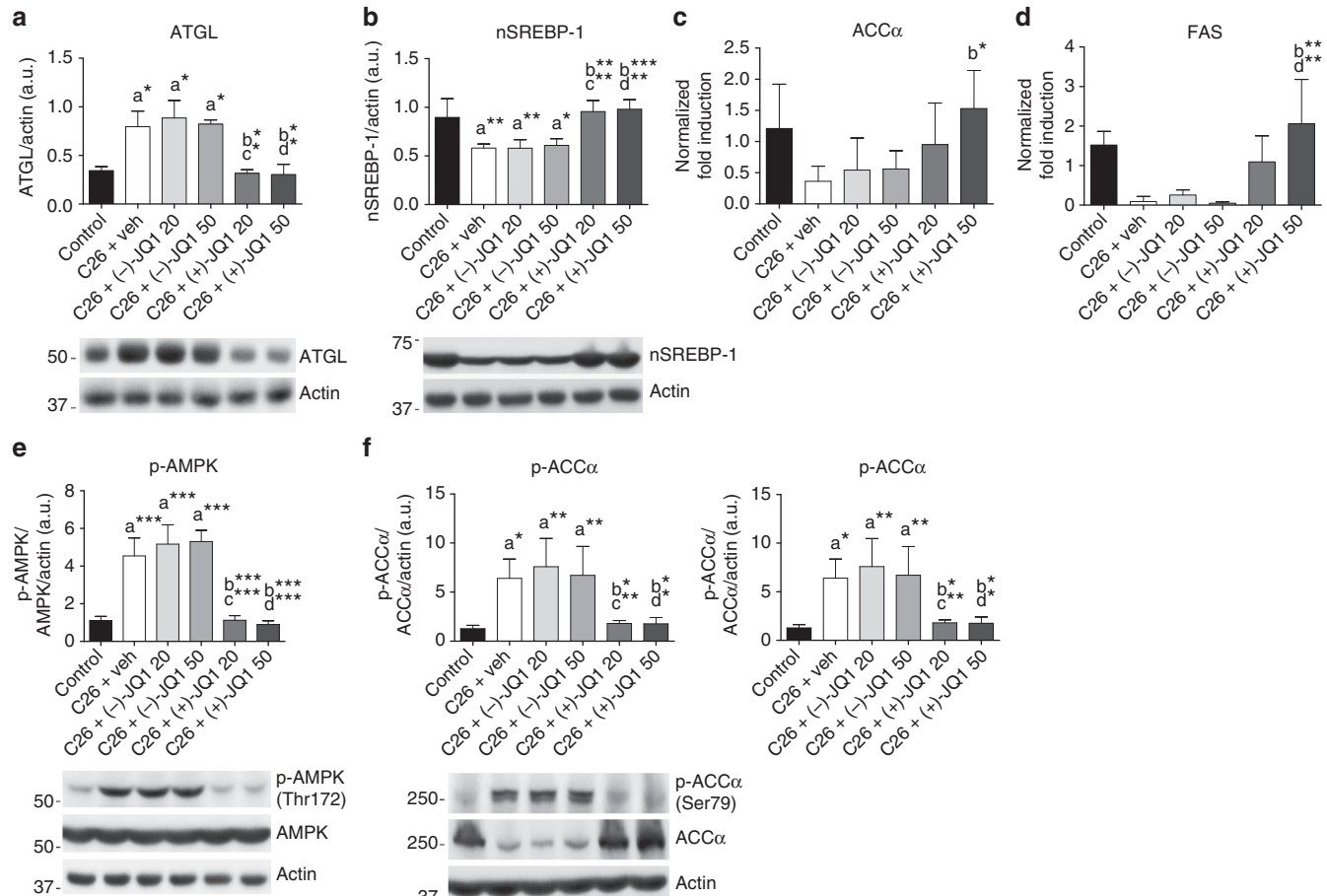

**Fig. 8** JQ1 administration blocks epididymal adipose tissue wasting. **a** Representative western blot and densitometric analysis of ATGL expression in epididymal adipose tissue (eAT) from control animals and C26-tumor-bearing mice treated with vehicle, (−)-JQ1, and (+)-JQ1 (animals per group: $n = 3$). Data represent means ± SD. **b** Representative western blot and densitometric analysis of the nuclear and transcriptionally active fragment of SREBP-1 (nSREBP-1) in the eAT of control and C26-tumor-bearing mice treated with vehicle, (−)-JQ1, and (+)-JQ1. Four animals were used for each experimental condition. Data represent means ± SD. **c, d** Total RNA was extracted from eAT of control and C26-tumor-bearing mice (animals per group: $n = 5$) treated with vehicle or JQ1 (−/+) and expression levels of ACCα and FAS were measured by quantitative RT-PCR. Data represent means ± SD. **e** Representative western blot and densitometric analysis of p-AMPK (Thr172) in eAT of the six experimental groups described in **a**. Four mice were used for each experimental group. Data represent means ± SD. **f** Representative western blot and densitometric analysis of ACCα protein expression and p-ACCα inhibitory phosphorylation (Ser79) in eAT of control and C26 tumor-bearing mice (animals per group: $n = 4$) treated with vehicle or JQ1 (−/+). Data represent means ± SD. Statistical analysis was performed by using one-way ANOVA followed by Tukey's post hoc test. $*p < 0.05$; $**p < 0.01$; $***p < 0.001$. "a" indicates statistical significance compared to control; "b" indicates statistical significance compared to C26+ vehicle; "c" indicates statistical significance compared to C26+ (−)-JQ1 20 mg/kg/day; "d" indicates statistical significance compared to C26+ (−)-JQ1 50 mg/kg/day

diameter (Fig. 5g; Supplementary Fig. 9a). AMPK blockade by Compound C hampered IL6 pro-atrophic effects. Moreover, Compound C administration reduced IL6-induced FoxO3 (Ser413) phosphorylation and MAFbx/Atrogin-1 upregulation (Fig. 5h; Supplementary Fig. 9b).

To shed light on the in vivo molecular mechanisms underlying the AMPK/FoxO3 interplay, we analyzed AMPK(Thr172) and FoxO3(Ser413) phosphorylation levels in TA muscles from vehicle, (−)-JQ1 and (+)-JQ1 tumor-bearing mice. AMPK was strongly activated in muscles of C26 vehicle and (−)-JQ1-treated tumor-bearing mice; the increase in AMPK(Thr172) phosphorylation paralleled a concurrent dramatic increase in FoxO3 (Ser413) phosphorylation. Both p-AMPK(Thr172) and p-FoxO3 (Ser413) levels were comparable in muscles from (+)-JQ1-treated C26-tumor-bearing mice and control animals (Fig. 6a). We further investigated alternative mechanisms underlying FoxO3 modulation and asked whether the Akt/FoxO3 and myostatin/ Smad3 axis were affected by implanted C26 colon carcinoma cells and JQ1 administration. Total FoxO3 protein levels were

modestly increased in TA muscles from C26-tumor-bearing mice (Fig. 6a), in agreement with previous reports[44,45]. Akt-phosphorylated FoxO3(Ser253) was not affected by tumor implantation as well as by (+)-JQ1 treatment (Fig. 5a). Consistently, p-Akt(Ser473) was comparable in the four animal groups (Supplementary Fig. 10a). Myostatin and Smad3 phosphorylation were also similar in control, vehicle, and (+)-JQ1-treated muscles from C26-tumor-bearing mice (Supplementary Fig. 10b, c).

While Akt-mediated phosphorylation modulates FoxO3 cellular localization, AMPK-dependent phosphorylation appears to modulate FoxO3 transcriptional activation and to promote transcription of a subset of FoxO3 targets, as reported in mouse embryonic fibroblasts (MEFs)[41]. We asked whether increased AMPK-mediated FoxO3 phosphorylation affected FoxO3 cellular compartmentalization in skeletal muscle. p-FoxO3(Ser413) was primarily localized in the nucleus of control TA myofibers, and its levels considerably increased during cancer cachexia. Likewise, while being mainly cytoplasmic, the nuclear fraction of p-AMPK

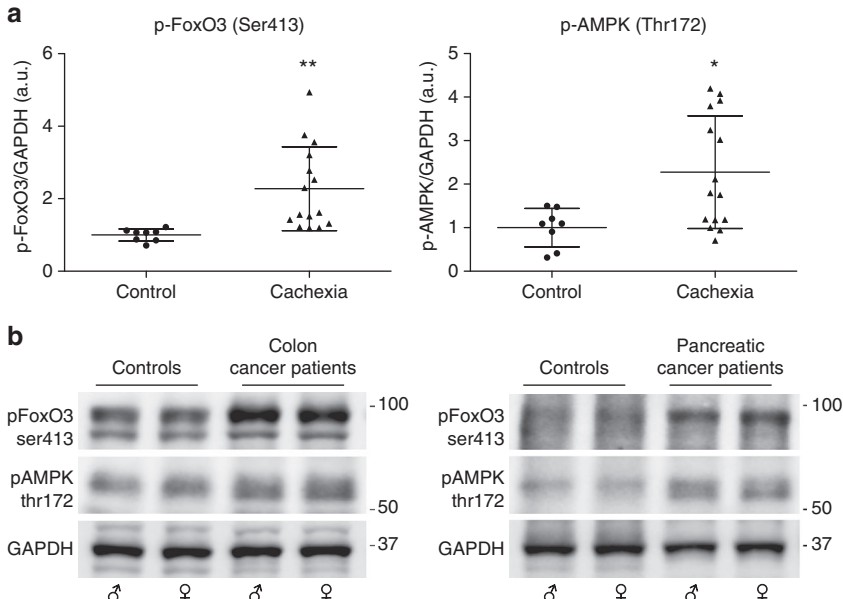

**Fig. 9** Cachectic cancer patients show increased levels of p-FoxO3Ser413 and p-AMPKThr172 in rectus abdominis muscle. **a** p-FoxO3 (Ser413) and p-AMPK (Thr172) levels were measured by immunoblot in rectus abdominis muscles from non-neoplastic patients ($n = 8$) and cachectic cancer patients ($n = 15$). GAPDH serves as loading control. Data are presented in scatter dot plot, lines at means $\pm$ SD. Statistical analysis was performed by using Student's unpaired $t$-test. *$p < 0.05$; **$p < 0.01$. **b** Representative western blots from non-neoplastic patients (controls), colon cancer, and pancreatic cancer patients are shown

(Thr172) augmented in muscles from cachectic mice and (+)-JQ1 treatment prevented p-AMPK(Thr172) and p-FoxO3(S413) nuclear accumulation (Fig. 6b). Immunostaining of TA muscle sections confirmed the nuclear localization of p-FoxO3(Ser413) (Fig. 6c) and p-AMPK(Thr172) (Supplementary Fig. 5d) in muscles from cachectic animals. Positive staining was limited to few spared nuclei in control mice, but dramatically increased in muscles from vehicle-treated C26-tumor-bearing mice. Both p-AMPK(Thr172) and p-FoxO3(Ser413) staining was reduced in (+)-JQI-treated animals (Fig. 6c; Supplementary Fig. 10d).

**AMPK and FoxO3 regulate the transcription of catabolic genes**. FoxOs transcription factors are a central node in skeletal muscle during cancer cachexia, coordinating gene networks that lead to muscle wasting[46]. Judge et al. (2014) identified a subgroup of FoxO-regulated genes that are upregulated in response to C26 tumor burden and are involved in muscle wasting[47] (Judge et al., 2014). We analyzed BRD4 engagement for this subset of FoxO-regulated genes at regulatory regions active in skeletal muscle, defined by the presence of DNAse I hypersensitive sites (DHS) (data obtained from Encode Project). BRD4 tag density distribution on DHS sites of this subgroup of FoxO-regulated genes revealed an increased BRD4 occupancy during cancer cachexia, which was abrogated by (+)-JQ1 treatment (Fig. 7a). These data indicate that BRD4 engagement increases at regulatory regions of a subgroup of wasting-related and FoxO-regulated genes during cachexia and that BRD4 association is challenged by (+)-JQ1. We therefore investigated FoxO3 recruitment to *MAFbx/Atrogin1*, *MuRF1*, and *GABARAPL1* promoters in skeletal muscles from control and (−)-JQ1 and (+)-JQ1-treated C26-tumor-bearing mice by ChIP qPCR. We observed that FoxO3 recruitment at these promoters was remarkably increased in muscles from tumor-bearing mice, while FoxO3 occupancy at these sites was similar in muscles from control and (+)-JQ1-treated tumor-bearing mice (Fig. 7b). Taking into account that p-AMPK(Thr172) increased nuclear localization during cachexia, we decided to elucidate whether nuclear

p-AMPK(Thr172) may be recruited on the chromatin, and we also queried promoter regions of pro-atrophy genes by ChIP. We observed that p-AMPK(Thr172) occupies the *MAFbx/Atrogin1*, *MuRF1*, and *GABARAPL1* promoters and that its association increased during cachexia. Conversely, (+)-JQ1 administration to C26-tumor-bearing mice led to a reduced association of p-AMPK (Thr172) at these chromatin regions (Fig. 7c).

To investigate whether AMPK-dependent FoxO3 phosphorylation plays a role in FoxO3 transcriptional activation, we employed a FoxO3 mutant in which the AMPK-phosphorylated residues are mutated to alanines (FoxO3-6A). Myogenic C2C12 cells stably overexpressing FoxO3-WT or FoxO3-6A were differentiated for 3 days in differentiation medium to form myotubes, which were then starved in low glucose medium for 24 h. Immunofluorescence assays showed that the six mutations do not affect FoxO3's ability to translocate to the nucleus upon starvation (Supplementary Fig. 11a–c). Foxo3-WT overexpressing cells had a significantly ($p < 0.0001$, one-way ANOVA) reduced diameter following starvation, while FoxO3-6A overexpressing myotubes were protected from atrophy (Fig. 7d; Supplementary Fig. 11d). In contrast to FoxO3-WT overexpressing cells, FoxO3-6A overexpressing myotubes failed to up-regulate MuRF1 and GABARAPL1 transcripts during starvation (Fig. 7e). Likewise, treating differentiated FoxO3-WT and FoxO3-6A overexpressing myotubes with conditioned medium from C26 cells (C26-CM) showed that FoxO3-6A overexpressing cells were resistant to C26-CM induced myotubes atrophy (Supplementary Fig. 11e).

Overall, these data suggest that phosphorylation of these sites plays a crucial role in FoxO3 activation during pro-atrophic conditions.

**JQ1 treatment spares epididymal adipose tissue wasting**. To assess whether the intraperitoneal administration of BET inhibitors impacted other tissues in a more systemic manner, we analyzed epididymal adipose tissue, which together with skeletal muscle was spared by (+)-JQ1 administration (Fig. 2d). In response to circulating inflammatory cytokines, the levels of

lipolytic enzymes increase to mobilize lipids and induce fat depletion during cancer cachexia[47,48]. (+)-JQ1 treatment prevented the upregulation of the lipase ATGL occurring in epididymal adipose tissue of vehicle and (−)-JQ1-treated C26-tumor-bearing mice (Fig. 8a). Moreover, BET blockade by (+)-JQ1 favored the activating cleavage of the lipogenic transcription factor sterol regulatory element-binding protein 1 (SREBP-1), which was suppressed in adipose tissue from cachectic mice (Fig. 8b). SREBP-1 promotes the expression of transcriptional targets as acetyl CoA carboxylase (ACC1) and fatty acid synthase (FASN), to stimulate fatty-acid synthesis[49]. Accordingly, ACC1 and FASN transcripts decreased in vehicle and (−)-JQ1-treated tumor-bearing mice. Transcript levels were comparable in control and (+)-JQ1-treated tumor-bearing mice (Fig. 8c, d). Because of AMPK crucial role as a metabolic sensor driving lipolysis, lipid mobilization, and biosynthesis[39,49,50], we investigated AMPK activation in epididymal adipose tissue in the six animal groups. AMPK(Thr172) phosphorylation dramatically increased in C26-tumor-bearing mice[38]. Similarly to skeletal muscle, (+)-JQ1 administration prevented p-AMPK(Thr172) increase in white fat (Fig. 8e). AMPK phosphorylates the rate-limiting enzyme in fatty-acid synthesis acetyl CoA carboxylase alpha (ACCα), which is consequently converted to an inactive form. In agreement with these findings, AMPK-dependent ACCα phosphorylation increased in C26-tumor-bearing mice treated either with vehicle or (−)-JQ1, but was maintained at control levels in (+)-JQ1-treated C26-tumor-bearing mice. Moreover, ACCα total protein levels decreased in vehicle and (−)-JQ1-treated mice, but not upon (+)-JQ1 treatment, in agreement with the analysis of SREBP-1 active form and ACC transcript (Fig. 8f). Collectively, these data suggest that (+)-JQ1 administration impairs white fat loss in cachexia through the modulation of AMPK and key actors in the lipogenic and lipolytic pathways.

**p-AMPK and p-FoxO3 increase in muscles of cachectic patients**. To evaluate whether the levels of p-AMPK(Thr172) and p-FoxO3(Ser413) are upregulated in skeletal muscle from cancer cachexia patients, we obtained muscle biopsies from cachectic patients affected by malignancies (colorectal and pancreatic cancers) in which cachexia has a 80% prevalence. Cancer patients were designated as cachectic according to the definition by Fearon et al.[51]. Immunoblot analysis of muscle biopsies revealed that both p-FoxO3(Ser413) ($p = 0.0058$, unpaired $t$-test) and p-AMPK(Thr172) ($p = 0.014$, unpaired $t$-test) levels were significantly higher in rectus abdominis muscle biopsies of cachectic cancer patients than in age-matched non-neoplastic patients (Fig. 9a, b; Supplementary Table 1). The results were comparable between the two cancer types and no sex-related differences were detectable, suggesting that the activation of AMPK/FoxO3 axis in skeletal muscle is a common event in cachectic cancer patients.

**Discussion**
Previous reports[5,6] and findings presented in this article highlight that preventing muscle and adipose tissue loss is a key element to prolong survival in experimental cancer cachexia models. Thus, development of therapies targeting cachexia should be a major goal to enhance cancer patients' quality of life, response to chemotherapy, and lifespan extension.

In this study, we pointed at BET blockade as a potential therapeutic approach to counteract cancer cachexia through two coordinated mechanisms, which directly target key players controlling muscle atrophy and pro-cachectic factors expression (Fig. 10).

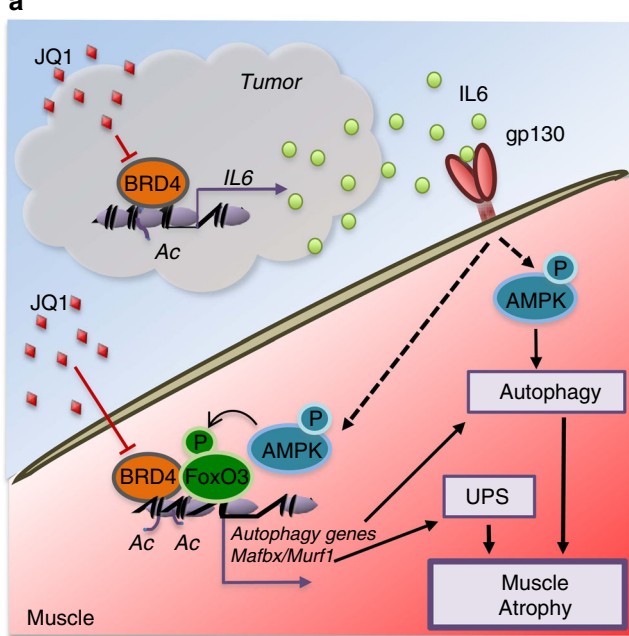

**Fig. 10** Proposed mechanism for BRD4 and JQ1 regulation of muscle atrophy during cachexia. **a** Schematic presentation of BRD4 functions in tumor and muscle, during cancer cachexia

JQ1 derivatives are currently in clinical trials as antineoplastic agents and show early encouraging results with leukemia and lymphoma patients[52]. Nevertheless, innate or induced resistance to BET inhibitors has been described[53–56] and novel combinatorial pharmacological approaches are developed. These approaches build on the mechanistic rationale that BET inhibitors resistance derives from acquired bromodomain-independent functions and by increased BRD4 phosphorylation levels. In this scenario, our results highlight that BET inhibitors may result beneficial independently of their direct antineoplastic activity, owing to their ability to restrain the expression of catabolic factors in skeletal muscle tissue.

At the onset of cancer cachexia, direct BRD4 and BRD2 recruitment at muscle catabolic genes is likely favored by local increase in histone acetylation, in concert with sustained engagement of transcription factors (e.g., FoxO3) and co-factors at chromatin regulatory regions. FoxO3 was shown to increase histone acetylation at its target loci, in DLD1-F3 cells overexpressing a constitutively active form of FoxO3[57,58]. We speculate that increased FoxO3 recruitment and histone acetylation promote BRD4 association, which in turn favors RNA PolII-mediated transcription. Interestingly, in a subset of genes such as *MuRF1* and *MAFbx/Atrogin-1* (Fig. 3), BRD4 also occupies regions of the gene body in muscles from C26-tumor-bearing mice, as previously reported for BRD4 targets in other experimental models[59]. Notably, FoxO transcription factors are acetylated at multiple lysine residues and this could represent an additional means of direct BRD4 and BRD2 recruitment, as previously described for the p65 subunit of NFkB, Twist[60], and STAT3[30]. We propose that in cancer cachexia, increased FoxO3 occupancy at pro-atrophic genes *MuRF1*, *MAFbx/Atrogin-1*, and *GABARAPL1* is promoted by AMPK-mediated FoxO3 phosphorylation, which is ultimately induced by increased systemic IL6 levels and AMPK activation in skeletal muscle (Fig. 10). Beside local effects on skeletal muscle, systemic levels of pro-cachectic modulators are also affected by (+)-JQ1 treatment. We showed a direct involvement for BRD4 and BRD2 in IL6 and

PTHrP transcriptional regulation in C26 tumors. IL6 and PTHrP appear to promote cancer cachexia through non-overlapping pathways, with a prominent role for PTHrP in hypercalcemia[33] and adipose tissue wasting[61]. Following (+)-JQ1 treatment, IL6 and PTHrP expression is reduced in tumors derived from C26 cells, suggesting that (+)-JQ1 modifies the tumor transcriptional program and prevents pro-cachectic factors upregulation, without affecting tumor mass. Notably, by impacting two or more mediators of cancer cachexia, (+)-JQ1 treatment may hamper adipose and muscle tissue wasting through distinct, but coordinated mechanisms.

Importantly, the systemic modulator IL6 has an impact on local muscular events through the IL6/AMPK/FoxO3 axis and likewise on other districts, such as adipose tissue. Because of the multifaceted nature of the disease and IL6 multisystemic functions, it is challenging to estimate the weight of local vs. systemic factors modulation. In this scenario, tempering multiple BET-regulated targets in different districts may be the key for an effective attenuation of cachexia symptoms.

We have focused our attention on pro-cachectic factors secreted by the tumor; nevertheless, systemic (+)-JQ1 delivery may concurrently reduce the transcription of pro-cachectic factors in other tissues and districts of the host. Cancer cachexia is a multi-organ metabolic syndrome, and since (+)-JQ1 is systemically delivered, its potential impact on tumor, inflammatory cells, adipose tissue, and skeletal muscle may synergistically be involved in cancer cachexia blockade.

We revealed a direct role for BET proteins both in skeletal muscle and in the tumor, by showing direct recruitment of BRD4 and BRD2 at a subset of pro-atrophic genes and pro-cachectic mediators.

Several reports link increased IL6 levels with AMPK activation[38,42,43]. Here we show that IL6-dependent AMPK activation signals to the nucleus and translates into FoxO3-dependent transactivation of catabolic genes. Notably, we discovered that AMPK is directly recruited to regulatory regions of catabolic genes. Compelling lines of evidence have shown that AMPK translocates to the nucleus, in several physiological and pathological conditions[62,63]. AMPK takes part in metabolism reprogramming, not only through direct regulation of metabolic enzymes, but also by influencing transcriptional regulation. AMPK phosphorylates a number of transcription factors and chromatin factors[64,65], as well as histone H2B at lysine 36, and directly associates to chromatin of stress-induced genes in MEFs cells[66]. Our findings reveal a key role for AMPK recruitment at chromatin regions of muscle catabolic genes, supporting the relevance of AMPK nuclear and chromatin functions in metabolism regulation. Importantly, we show that AMPK activation and AMPK-dependent FoxO3 phosphorylation are upregulated in cachectic cancer patients, hinting for a conserved role of AMPK/FoxO3 interplay in cancer cachexia.

Overall, BET inhibitors represent a promising strategy to coordinately counteract muscle wasting, by decreasing inflammation and by preventing activation of the muscle catabolic program.

## Methods

**Study design.** The pre-specified objective of the research was to evaluate whether BETs inhibition with the small molecule (+)-JQ1 was able to counteract cancer cachexia. This pre-specified hypothesis led us: (i) to investigate BRD4 contribution to muscle atrophy secondary to cancer and BRD4 role in regulating pro-cachectic factors levels and (ii) to dissect BRD4 interplay with FoxO3 and IL6/AMPK signaling in skeletal muscle. We studied experimental models of cancer cachexia, patients' muscle biopsies and cell culture assays.

A priori determination of sample size was calculated by analyzing the published data[26] and by evaluating the number of animals per group used for similar

experiments. Prior knowledge of the growth rate and variability of tumor models was also considered for the establishment of sample size.

The minimal sample size for morphological, western blot, and ChIP analysis was generally 3–5 animals per group, whereas functional and mRNA evaluations were assessed on 8–10 animals per group. For all in vitro data, the number of experiments is defined in the figure legend. In vitro experiments were independently repeated at least three times. In order to minimize the intragroup biological variation, all the experiments were performed in the same physiological conditions, by using 8-week-old male mice.

Final endpoints were determined prospectively and were based on the degree of cachexia and tumor volume of the vehicle-treated tumor-bearing mice. Exclusion criteria for animals were applied in case of death, cannibalism, and the presence of severe clinical alteration of vital physiological functions and ulceration induced by the tumor mass. Exclusion criteria for samples were applied in case of histological artifacts (freeze- and cut-damaged tissues), RNA and protein degradation, assessed by gel electrophoresis. Mice purchased from Charles River were randomly allocated to six different experimental groups. In vivo experiments and animal manipulation were not blinded. However, blind evaluation was performed for tissue weight, morphological analysis, mRNA estimation, and protein quantification of mouse-derived tissue samples. Statistical analysis was conducted upon verification of the Normality assumption by using the Kolmogorov–Smirnov test (where applicable). The similarity of variance among the experimental groups was also verified.

**Cell culture and materials.** MAC16, C26, and B16 cells were a kind gift of Drs Tisdale and Russell (School of Life & Health Sciences, Aston University Burmingham, UK), Professor Mario P. Colombo (IRCCS National Cancer Institute, Milano, Italy), and Professor F. Cavallo (Department of Molecular Biotechnology and Health Science, University of Turin, Italy), respectively. PC-3M and A375M were obtained from MD Anderson Cancer Center, MKN-1 from Riken BioResource Center (Japan), LLC, 4T1, C32, and A2058 cells were from ATCC. Cell lines were assessed monthly for Mycoplasma contamination with MyocAlert Mycoplasma detection kit (Euroclone, Italy).

MAC16, 4T1, A375M, MKN-1, and PC-3M cells were cultured in RPMI (Sigma, Italy), whereas C26, LLC, B16, and A2058 cells were cultured in high-glucose Dulbecco's Modified Eagle's medium (DMEM; Euroclone, Italy). All culture media were supplemented with 10% fetal bovine serum (Euroclone, Italy) and Pen/Strep solution (penicillin 100 U/ml and streptomycin 0.1 mg/ml) (Lonza, Euroclone, Italy) and maintained at 37 °C with a humidified atmosphere of 5% $CO_2$. Cell sensitivity to (+)-JQ1 was evaluated by seeding 100,000 cells in 35 mm dishes; each cell line was then treated with three different doses of (+)-JQ1 for 72 h. After 72 h, the cells were colored with trypan blue and viable cells were counted, to evaluate cell viability. Cell viability was represented showing the percentage of viable cells (control cell = 100% viability). C26 were treated with different concentration of (+)-JQ1 or (−)-JQ1 for 48 h before RNA extraction.

C2C12 myoblast cells (ATCC) were cultured in DMEM without Na-Pyruvate (Sigma, Italy) supplemented with 20% FBS, 2 mM L-glutamine, and 100 U/ml penicillin and 0.1 mg/ml streptomycin (Euroclone, Italy) at 37 °C and 5% $CO_2$ and induced to differentiate by switching to DM medium (DMEM supplemented with 2% horse serum) for 4 days.

Starvation was induced by switching myotubes in DMEM with 2.5 mM glucose (Invitrogen, Italy) and 0.1% horse serum (Invitrogen, Italy).

C2C12 were treated with 20 ng/ml rIL-6 (Abcam, UK) and/or 10 μM Compound C (Abcam, UK) diluted in DM medium.

**Lentivirus production and plasmids.** Lentiviral vectors were produced by transient co-transfection of pLKO, psPAX2 (#12260 Addgene), and VSVG (#8454 Addgene) plasmids into HEK293T. pLKO.1 plasmid targeting murin BRD4 was kindly provided from Dr. Hernando. pBabe-FoxO3WT and pBABE-FoxO3-6A were transferred from pECE-Flag-FoxO3-WT (#8360 Addgene) and pCDNa-Flag-FoxO3-6A (#24382 Addgene) in pCRII-TOPO by using TOPO TA Cloning kit (Life Technologies, Italy), and then sublconed in pBabe retroviral vector, through SalI digestion.

Retroviruses were produced by transient co-transfection of pCL-ECO (plasmid #12371, Addgene) and pBABE-Empty (plasmid #1764, Addgene) or pBABE-FoxO3-WT or pBABE-FoxO3-6A in HEK293T cells.

Virus was harvested 48 h post transfection and infections were carried out by spinoculation at 3000 rpm for 90 min, in presence of 4 μg/ml polybrene (Santa Cruz). Infected cells were selected with 4 μg/ml (C26 cell line) or 2 μg/ml (C2C12 cell line) puromycin (Sigma, Italy) for a week.

**RT-PCR and real-time PCR.** Total RNA from either snap-frozen tissues or cells was isolated using Trizol (Sigma, Italy) according to the manufacturer's instructions. After DNAse treatment (Ambion, Life Technologies, Italy), RNA was purified with RNA clean up Kit (Zymo, Italy), reverse transcribed to cDNA with the High-Capacity cDNA Reverse Transcription Kit (Applied Bio-System), and subjected to qPCR analysis. Primers were designed to amplify regions of 80–120 bp. Oligonucleotides sequences used in qRT-PCR are reported in Supplementary Table 2. Quantitative PCR was performed in triplicates using SYBR green IQ reagent (Bio-Rad Laboratories, Italy), with CFX Connect detection system (Bio-Rad Laboratories, Italy).

**ChIP and ChIP-seq**. Chromatin isolated from muscles or tumors was subjected to ChIP and ChIP-seq assays according to Savic et al.[67]. Briefly, frozen tissues were pulverized in Covaris tissue TUBEs (Covaris 520001) using a chilled hammer on a cold metallic block, in dry ice. Powdered tissue was transferred in a glass, where it was resuspended in PBS, cross-linked in 1% formaldehyde (Sigma, Italy) for 15 min. After blocking cross-link with 0.125 M glycine for 5 min and washes with PBS plus protease and phosphatase inhibitors, pellet was resuspended in Farnham buffer (5 mM PIPES pH 8.0; 85 mM KCl; 0.5% NP-40) and briefly homogenized with a tissue homogenizer.

Cells were subsequently lysed in RIPA buffer (1× PBS; 1% NP-40; 0.5% sodium deoxycholate; 0.1% SDS). For each antibody,100 mg of starting tissue was used. Chromatin was sonicated to fragments length of ~0.5 kb and immunoprecipitated with 3.5 μg of rabbit IgG or antibodies listed in Supplementary Table 4. Immunoprecipitation procedures were performed as in Proserpio et al.[17]. ChIP primers are listed in Supplementary Table 3. Quantitative real-time PCR was performed using SYBR green IQ reagent (Bio-Rad Laboratories, Italy) with CFX Connect detection system (Bio-Rad Laboratories, Italy). For ChIP-seq, 10 ng immunoprecipitated DNA fragments were used to prepare libraries with the NEBNext RNA Library Prep Kit (New England Biolabs) and the Ovation SP Ultralow DR Multiplex System (NuGEN), following the manufacturer's protocol. Libraries were sequenced for 50 cycles on a HiSeq 2000 or HiSeq2500 Illumina instrument. Peaks distribution was plotted using PAVIS and Gene Ontology analysis was performed using DAVID 6.8 bioinformatic resource.

**Animals experimental design**. Eight-week-old male BALB/c mice (Charles River, Italy) were housed in groups of five and maintained under controlled temperature ($20 \pm 1$ °C), humidity ($55 \pm 10\%$), and illumination (12/12 h light cycle with lights on at 07:30 am). Food and water were provided ad libitum. All mice were held in quarantine for 2 weeks before the experiments. Tubes for tunneling and nesting materials (paper towels) were routinely placed in all cages as environmental enrichment. All procedures involving animal care or treatments were approved by the Italian Ministry of Health and performed in compliance with the guidelines of the Italian Ministry of Health (according to the Legislative Decree 116/92, the Directive 2010/63/EU of the European Parliament and the Council of 22 September 2010 on the protection of animals used for scientific purposes.

Tumor-bearing mice received $10^5$ C26 colon carcinoma cells in PBS by dorsal s. c. injection as previously reported[15]. Control mice were inoculated with PBS. Animals were then randomized and divided into six groups: control (mice without tumor-cell inoculation, treated with vehicle), C26 + Veh (C26-bearing mice treated with vehicle), C26+(−)-JQ1-20 (C26-bearing mice, 20 mg/kg body weight of the inactive JQ1 enantiomer), C26+(−)-JQ1-50 (C26-bearing mice, 50 mg/kg body weight of the inactive JQ1 enantiomer), C26+(+)-JQ1-20 (C26-bearing mice, 20 mg/kg body weight of the active JQ1 enantiomer), and C26+(+)-JQ1-50 (C26-bearing mice, 50 mg/kg body weight of the active JQ1 enantiomer). Active and inactive JQ1 enantiomers were dissolved in vehicle (10% 2-hydroxypropyl-β-cyclodextrin, 10% DMSO) at a final concentration of 5 mg/kg, and daily administered by i.p. injections during the whole experimental period. Animal weight and food intake were recorded daily. Twelve days after C26 implantation, the mice were deeply anesthetized using tribromoethanol (250 mg/kg) and the blood was collected into EDTA (1 mg/ml blood) for plasma analyses. Subsequently, mice were decapitated and their tissues quickly collected. In particular, skeletal muscles, adipose tissue, and C26 tumors were excised, weighed, and frozen in cooled isopentane with liquid nitrogen and stored at −80 °C for subsequent biochemical and morphological analyses. In survival studies, when treatment was started at moderate cachexia, (+)-JQ1 (20 mg/kg body weight) or vehicle administration began when C26-tumor-bearing mice average weight loss was 10%. Animal weight and food intake were recorded daily. For B16 melanoma-induced cachexia, a suspension of $10^5$ B16 cells in PBS was subcutaneously injected in C57BL/6 mice. Ten days after tumor-cell inoculation, the animals received a daily dose of (+)-JQ1 (20 mg/kg) or vehicle (10% 2-hydroxypropyl-β-cyclodextrin, 10% DMSO). The mice were sacrificed 3 weeks after tumor implantation.

**Treadmill test**. Animals were subjected to an exercise tolerance test every week, starting from the initial day of tumor cell inoculation and JQ1 treatment. Briefly, the mice were acclimatized to treadmill running with a 10-min run at a constant speed of 6 m/min for five consecutive days prior to the first exercise. During the test sessions, mice were run at an initial speed of 6 m/min, and every 2 min speed was increased by 2 m/min until exhaustion. The animals were encouraged to run with an electrical shock grid at the back of the treadmill (1.5 mA, 200 ms pulses, 4 Hz). The study was performed until the mouse was exhausted as defined by remaining on the electrified grid for at least 5 s. The first exercise test was used to set the baseline of each experimental group.

**Interleukin-6 detection**. Plasma IL-6 concentration was measured on plasma samples by ELISA kit (Invitrogen Life Technologies, Italy) according to the manufacturer's instructions.

**In vitro protein degradation assay**. Fast MyHC degradation was assayed according to a previously described protocol[68]. Briefly, TA samples were

homogenized in ice cold 0.01 M Tris-HCl (pH 7.4), 0.150 M sucrose. Total protein concentration was performed by the method of Lowry et al.[68]. Homogenate (30 μg) was used for each reaction. Samples were incubated at 37 °C, the incubation was blocked by adding an equal volume of sample buffer (0.125 M Tris-HCl containing 10% SDS, PMSF, protease inhibitor cocktail, pH 6.8), at different time points. The samples were boiled for 3 min and loaded onto acrylamide gel for western blot analysis.

**Western blot**. For total lysate preparation, mouse tissues (TA muscles, epididymal adipose tissue, and C26 tumors) and cultured cells were homogenized in a homogenization buffer (0.01 M Tris-HCl, 0.001 M CaCl2, 0.15 M NaCl, 1 mM PMSF, protease inhibitor cocktail, and phosphatase inhibitor cocktail, pH 7.5). The homogenate was then lysed by sonication. Cytosolic and nuclear fractionation was prepared as in Dimauro et al.[69]. Protein concentration was assessed by the method of Lowry et al.[68]. Samples were boiled for 3 min before loading to the SDS-PAGE. Proteins (30 μg) were resolved by 12% (for caspase-3, LC3b, interleukin-6, H3, and PTHrP), 10% (for Myc, Atrogin-1, Beclin-1, AMPK, Akt, Myostatin, ATGL, and SMAD3), and 7% (for MyHC fast, MyHC slow, Ulk, FoxO1, FoxO3, STAT3, SREBP-1, and ACC), and SDS-PAGE at 30 mA (constant current) for 60 min. Proteins were transferred onto nitrocellulose membrane using the trans-blot turbo transfer system (Bio-Rad Laboratories, Italy) for 10 min at room temperature. The nitrocellulose membrane was blocked with 5% fat-free milk or 3% BSA in Tris-buffered saline (0.138 M NaCl, 0.027 M KCl, 0.025 M Tris-HCl, and 0.05% Tween-20, pH 6.8) at room temperature, and probed at 4 °C overnight with primary antibodies followed by incubation for 1 h with horseradish peroxidase-conjugated secondary IgG antibodies (Bio-Rad Laboratories, Italy). The nitrocellulose membrane was then re-probed with anti-Vinculin, anti-GAPDH, anti-Actin, or anti-H3 antibodies. Bound antibodies were visualized using Clarity Western ECL substrate (Bio-Rad Laboratories, Italy) and image acquisition was performed through ChemiDoc MP system (Bio-Rad Laboratories, Italy). Uncropped images of most relevant immunoblots are provided in Supplementary Fig. 12. Images derived from western blot were analyzed with ImageJ (National Institute of Health, Bethesda, MD, USA) software. All samples were normalized for protein loading by using Vinculin, GAPDH, Actin, or H3 (chosen as housekeeping proteins). Recorded value was obtained from the ratio between arbitrary units derived by the protein band and the respective housekeeping protein. All primary antibodies used in the Western blot procedure are listed in Supplementary Table 4.

**Morphological analysis**. Immunostaining of cultured cells were carried out as described previously[17]. Formaldehyde fixed cells were incubated with appropriate antibodies and DAPI was used for nuclear staining. The samples were examined with a fluorescence microscope (Carl Zeiss, Italy). Pictures of staining were obtained using an AxioCam (Carl Zeiss Vision, Italy). Antibodies are listed in Supplementary Table 4.

In vivo morphological evaluation was performed on OCT frozen TA muscles. Transverse, 10-μm-thick sections were cut by a cryostat and collected on Superfrost Plus slides (BioOptica). For each TA muscle ($n = 3$ per experimental group), a minimum of 20 sections were processed for hematoxylin–eosin (H&E) staining, dehydrated and mounted with Eukitt (Kindler GmbH & Co., Germany). For immunofluorescence analysis of TA muscles, 10-μm frozen sections were blocked with 10% Normal Goat Serum (NGS, Vector Laboratories) in PBS with 2.5% Triton-X 100, and then incubated overnight at 4 °C with anti-p-AMPK (1:50, anti-p-AMPK (Thr172)) and anti-p-FoxO3 (1:20, anti-p-FoxO3(Ser413)). Negative controls were performed omitting the primary antibodies. Sections were thoroughly rinsed with PBS, then incubated for 1 h at room temperature with PBS containing 5% NGS and Alexa488-conjugated goat anti-rabbit IgG (1:500, Invitrogen Life Technologies, Carlsbad, CA, USA). Slides were mounted with Fluoroshield mounting medium with DAPI (Sigma Aldrich, Italy). Muscle slices were observed under a Leica CTR6000 microscope (Leica, Germany) equipped with Leica DFC360 camera (immunofluorescence visualization) and Leica DFC480 (bright-field visualization). Images were captured using Leica Application Suite System and files were converted in Adobe photoshop CS5 format.

**Patients' skeletal muscle samples**. Muscle samples were collected from rectus abdominis muscle of 15 cachectic tumor patients (3 males colorectal cancer, 3 females colorectal cancer, 5 male pancreatic cancer, and 4 females pancreatic cancer patients) and 8 non-tumor patients (3 males and 5 females). All patients underwent elective laparoscopic surgery, during which muscle biopsy was performed. Muscle biopsies were snap frozen in liquid nitrogen immediately after surgery.

For each patient, a small piece of frozen muscle biopsy was powdered and lysed in a buffer containing 50 mM Tris, pH 7.5, 150 mM NaCl, 5 mM MgCl2, 1 mM DTT, 10% glycerol, 1% SDS, 1% Triton X-100, 1X Roche Complete Protease Inhibitor Cocktail, 1× Sigma-Aldrich Phosphatase Inhibitor Cocktail 1 and 3. Then, samples were immunoblotted and visualized with SuperSignal West Pico Chemiluminescent substrate (Pierce). Blots were stripped using Restore Western Blotting Stripping Buffer (Pierce) according to the manufacturer's instructions and were reprobed if necessary.

All enrolled subjects were volunteers who signed an informed consent to participate in the study that took place at the Department of Surgery, Oncology and Gastroenterology, 3rd Surgical Clinic, University of Padua. Study protocol (Protocol 3674/AO/15) was approved by Ethical Committee for Clinical Research in the district Padova (CESC).

Sample acquisition and preparation, data handling and encryption were performed according to the 1996 Declaration of Helsinki.

**Statistical analysis**. Data obtained from functional, morphological, western blot, and mRNA analysis are expressed as means ± SD (standard deviation), whereas data obtained from ChIP experiments are represented as means ± SEM (standard error of the mean). When we compared two experimental groups, we used unpaired Student's $t$-test and when we compared three or more experimental groups, we used one-way analysis of variance (ANOVA) followed by the Tukey's post hoc test. Values of $p < 0.05$ were considered to indicate a significant difference. Statistical analysis was performed using GRAPHPAD INSTAT3 (GraphPad, La Jolla, CA, USA) for Windows.

**Data availability**. The BRD4 ChIP-seq data have been deposited in the GEO database under the accession code GSE104155. The authors declare that all the other data supporting the findings of this study are available within the article and its Supplementary Information files and from the corresponding author, upon reasonable request.

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

## Acknowledgements

We are sincerely grateful to the patients who participated in our study. We are thankful to Dr L.P. Pucillo for cholesterol blood analysis, Drs M. Mora and S. Gibertini for help with muscle sectioning, Drs G. Gutierrez-Cruz and S. Dell'Orso for NGS sequencing, Dr G. Messina's group for support with the treadmill test, Drs Hernando and Di Micco for sharing BRD4 Sh-RNA plasmids, and Dr M.J. Tisdale and S.T. Russell for sharing MAC-16 cells. This work was supported by grants from Worldwide Cancer Research (14-0149), and, in part, by Telethon GGP13165, French Association for Myophaties (AFM-Telethon), AIRC-Trideo to G.C., by the Intramural Research Program of the NIAMS at the NIH to V.S. and by the Wellcome Career Development Fellowship (095751/Z/11/Z) to P.F. R.F. is supported by a FIRC fellowship.

## Author contributions

M.Se. performed animal treatments, immunoblots, immunohistochemical, morphological analysis, helped with C2C12 experiments, designed experiments, analyzed the data, and was involved in manuscript preparation. R.F. performed RNA analysis, C2C12 and C26 experiments, and helped with animal treatments. C.F. performed ELISA experiments. R.S., Z.G., P.E.S., S.C., S.M., and M.Sa. collected human samples; R.S., M.Sa., and M.Se. performed immunoblots and analyzed the data on patients' samples. P.F. provided JQ1 and contributed in experimental design. F.P. and P.C. performed in vivo experiments with B16-tumor-bearing mice. H.Z. and K.D.K. performed bioinformatics analysis of ChIP-seq data. S.H. helped to set up in vivo experiments. V.S. contributed in experimental design and manuscript preparation. G.C. conceived the study and designed experiments, performed tissue ChIP-seq and ChIP assays, analyzed the data, and wrote the manuscript. All authors discussed the results and commented on the manuscript.

## Additional information

**Competing interests:** S.H. is an employee of the Novartis Institutes of Biomedical Research. The remaining authors declare that they have no competing financial interests.

