## [Peer Review File · Nature Communications]

Reviewer #1:
(Remarks to the Author):

The current study extends the investigative teams prior work examining BRD4 in the regulation of myotubes size, and the use of a BET inhibitor to prevent dexamethasone induced atrophy. The stated purpose was to determine if BRD4 is involved in the transcriptional regulation of catabolic genes during cancer cachexia. Furthermore, the study examines the effects of a BET inhibitor on cachexia development using the C26 mouse model of cancer cachexia. The authors conclude that the BET inhibitor can block catabolic gene activity in muscle and link BRD4 to catabolic gene activation in muscle through IL-6 / AMPK/ FOXO3 signaling. This is a technically sound manuscript that goes to great lengths to examine the effect of systemic BRD4 inhibition of the development of cachexia in an established preclinical mouse model. A strength of the study is the detailed analysis to establish the effects of the inhibitor on weight loss. The study goes far beyond standard, lower impact studies in the field, and provides novel data on the inhibitor effects on tumor cells in vitro and in vivo. The study also quantifies food intake and exercise capacity, 2 variables left out of lower impact studies. There is also extensive novel analysis of BRD4 target genes during cachexia, and some clinical data in cancer patients.

Major Comments

Despite numerous strengths the impact of the study is severely diminished by the presentation that centers on a simple design involving inhibitor administration. These makes the study a somewhat less novel inhibitor study of cachexia, and the design has no further manipulations that provide mechanistic insight into the proposed mechanisms. The discussion does not adequately discuss the main finding of systemic versus local effects in enough detail or breadth.

Repositioning the presentation to focus on the strength of the study involving BRD4 in cachexia, and use the inhibitor as only one manipulation in a larger study would dramatically increase the overall impact of the study.

As a simple cachectic prevention design, most measured changes in variables found in muscle and fat are related to the fact that the inhibitor lowered circulating IL-6, maintained food intake, and muscle mass loss was mitigated. A weight stable muscle that does not activate cellular wasting signaling is not novel. This mechanistic deficit could be offset by many potential additional experiments, for example giving the inhibitor after weight loss as been initiated, examining pre-cachectic tumor bearing mice. A standard in the field is that the mechanistic examination of muscle wasting during cachexia includes treatments that do not affect the underlying disease, and experiments that can determine direct versus systemic effects on muscle signaling. The systemic the inhibitor centered study fails on both counts.

Specific

Introduction: A stronger premise in the introduction is needed. While it is mentioned in the manuscript several times that there are no approved treatments for cancer cachexia, a proposed reason for this fact is not presented, cachexia has been widely investigated for 25 years. Furthermore, how the current inhibitor study overcomes this gap is not clearly stated. IL-6, STAT3, E3 ligases, autophagy and even AMPK have been widely investigated as potential therapeutic targets.

Introduction: First sentence needs a reference.

Results: Why was the in vivo study stopped at 12 days when the animals survived another 5-7 days? What was the advantage examining 48-72 window of cachexia? Why were tumor bearing non-cachectic mice not included in the analysis 10 days? Why did the study not examine the

benefits of the inhibitor administered after the initiation of weight loss on survival, which is more clinically relevant to a therapeutic target?

Results: Based on the survival data can the reader assume that the inhibitor only delayed the onset of cachexia? Please provide a justification for your answer, as this would be of interest to those trying to interpret the findings.

Results: Is it surprising that catabolic pathways are not activated in a non-wasting muscle? More exciting would be catabolic pathways activated that no longer lead to wasting, do the authors have any information in this area?

Can the authors provide another AMPK downstream phosphorylation target other than FOXO, there are many and this would strengthen the interpretation?

Results, epididymal adipose section: If the author's feel this is an important point for the manuscript some of the adipose wasting data should be moved from the supplement to the actual paper, or discussion of this point minimized

Figure 8 data and text. There needs to be some descriptor data of the patients and weight loss. Please add in text or provide in supplement to help with interpretation.

Methods: Provide the N analyzed for each figure or assay in the legend.

Statistical analysis: related to rigor, please clarify where unpaired t-tests and ANOVA analysis were used.

Figure 5G & 5H. Long-term compound C administration has often been reported to be toxic to myotubes. Please include the Compound C only controls.

Figure 7D. Please clarify if insulin levels were controlled for. It is also of interest why defined cachectic media from tumor cells was not used in place of serum starvation. Please clarify.

Page 12, first sentence: an incomplete sentence

Reviewer #2

(Remarks to the Author):

The manuscript by Caretti and co-workers describes the positive effects of the pan-BET inhibitor JQ1 in a C-26 colon carcinoma model of cancer cachexia, preventing weight loss of tumor-bearing mice and reducing muscle and adipose tissue wasting. The effect is accompanied by increased survival rate of the animals in this model. The BET bromodomain BRD4 is at least in part responsible for this effect by regulating the cytokine IL-6, a well-known downstream target of BRD4, as well as through modulation of the AMPK pathway and prevention of phosphor-AMPK nuclear accumulation. Transcriptional regulation of FOXO3, a key transcription factor of catabolic pathways, was also affected by (+)-JQ1 treatment. Moreover treatment with (+)-JQ1 prevented UPS and autophagy induction muscles of C26-tumor bearing mice.

Based on their recent finding that BRD4 plays a role in controlling myotube size and that (+)-JQ1 was able to prevent atrophy of myotubes in a dexamethasone induced model the authors investigated if BRD4 is involved in transcriptional regulation of cachexia induced catabolic genes. ChipSeq experiments confirmed the association of BRD4 with regulatory sites in several catabolic genes. Treatment of mice with (+)-JQ1, but not the negative control compound (-)-JQ1 prevented the activation of catabolic genes associated with skeletal muscle atrophy and decreased IL6 systemic levels. They also show that in biopsies from cachectic patients these pathways are

relevant.

The finding that BRD4 plays a role in cachexia is new and the author provide a comprehensive analysis of the pathway involved and its regulation by BET bromodomains. Given the fact that BET inhibitors are already in several clinical trials in oncology this study is highly relevant and timely. I recommend the manuscript for publication pending some additional amendments:

- Abstract: (+)-JQ1 protects tumor-bearing mice from body weight loss, muscle and adipose tissue wasting. Remarkably (+)-JQ1 administration dramatically prolongs survival without directly affecting tumor growth. – This statement is slightly misleading as it is phrased, implicating that the lack of affect in tumor proliferation is a general effect. It should be rephrased in a way to make it clear that in a single cell line (+)-JQ1 did not affect tumor growth and that this cell line was used for further studies of cachexia.
- What is the method used for tumor cell antiproliferative effect in Fig 1? Which cancer is associated with the different cell lines used in Fig 1. Which method was used to measure proliferation
- Fig 2 treadmill test: why was the test not performed using (-)-JQ1 tested? Fig 2D: is there statistical significance between vehicle-treated animals and animals treated with (-)-JQ1. If so what is the rationale for this? Likewise in Figure 3A
- P.8 2nd paragraph; correct spelling of: '(+)-JQ1' restrains cancer cachexia through the reduction of IL6...
- BRD2, another Bet family protein is potently inhibited by (+)-JQ1 as well. BRD2 has been linked to metabolism with BRD2 loss in mice causing obesity. The author should show if BRD2 is involved in cachexia.

Reviewers' comments:

We would like to thank both reviewers for their overall positive assessments of our manuscript and for their detailed critical comments that we believe led to improvements in our manuscript and therefore enhanced its potential clinical implications.

Reviewer #1: Cancer cachexia

(Remarks to the Author):

The current study extends the investigative teams prior work examining BRD4 in the regulation of myotubes size, and the use of a BET inhibitor to prevent dexamethasone induced atrophy. The stated purpose was to determine if BRD4 is involved in the transcriptional regulation of catabolic genes during cancer cachexia. Furthermore, the study examines the effects of a BET inhibitor on cachexia development using the C26 mouse model of cancer cachexia. The authors conclude that the BEYT inhibitor can block catabolic gene activity in muscle and link BRD4 to catabolic gene activation in muscle through IL-6 / AMPK/ FOXO3 signaling. This is a technically sound manuscript that goes to great lengths to examine the effect of systemic BRD4 inhibition of the development of cachexia in an established preclinical mouse model. A strength of the study is the detailed analysis to establish the effects of the inhibitor on weight loss. The study goes far beyond standard, lower impact studies in the field, and provides novel data on the inhibitor effects on tumor cells in vitro and in vivo. The study also quantifies food intake and exercise capacity, 2 variables left out of lower impact studies. There is also extensive novel analysis of BRD4 target genes during cachexia, and some clinical data in cancer patients.

Major Comments

Despite numerous strengths the impact of the study is severely diminished by the presentation that centers on a simple design involving inhibitor administration. This makes the study a somewhat less novel inhibitor study of cachexia, and the design has no further manipulations that provide mechanistic insight into the proposed mechanisms.

Thank you for your encouraging review and your forthcoming critical comments. To provide additional mechanistic insight, we performed new ChIP assays on tissues from *in vivo* experiment. We have found that the BET protein BRD2 associate with Atrogin-1 and GABARAPL1 promoters, in tibialis anterior muscles, and at the IL6 and PTHrP promoters in tumors. Interestingly, BRD2 and BRD4 have been found to co-regulate other sets of targets, impacting cell cycle, immunity and pluripotency genes, with distinct but coordinated functions (see also recent reports^{1,2,3}).

This is a novel and mechanistic finding, which will require further characterization of BRD2/BRD4 interplay in terms of specificity, time sequence and potential synergism, but at the same time this observation makes genetic manipulation in mouse models focused only on BRD4 functionally less informative. From a therapeutic perspective, this observation amplifies the broad spectrum of action of BET inhibitors, which directly block both BRD4 and BRD2 bromodomains function, thus presenting promising therapeutic efficacy.

The discussion does not adequately discuss the main finding of systematic versus local effects in enough detail or breadth.

We have modified the discussion to address local versus systemic effects.

Please see page 14: “Beside local effects on skeletal muscle, systemic levels of pro-cachectic modulators are also affected by (+)-JQ1 treatment. We showed a direct involvement for BRD4 and BRD2 in IL6 and PTHrP transcriptional regulation in C26 tumors. IL6 and PTHrP appear to promote cancer cachexia through non-overlapping pathways, with a prominent role for PTHrP in hypercalcemia⁴ and adipose tissue wasting⁵. Following (+)-JQ1 treatment, IL6 and PTHrP expression is reduced in tumors derived from C26 cells, suggesting that (+)-JQ1 modifies the tumor transcriptional program and prevents pro-cachectic factors upregulation, without affecting tumor mass. Notably, by impacting two or more mediators of cancer cachexia, (+)-JQ1 treatment may hamper adipose and muscle tissue wasting through distinct but coordinated mechanisms. Importantly, the systemic modulator IL6 has an impact on local muscular events through the IL6/AMPK/FoxO3 axis and likewise on other districts, such as adipose tissue. Because of the multifaceted nature of the disease and IL6 multisystemic functions, it is challenging to estimate the weight of local versus systemic factors modulation. In this scenario, tempering multiple BET-regulated targets in different districts may be the key for an effective attenuation of cachexia symptoms. “

Repositioning the presentation to focus on the strength of the study involving BRD4 in cachexia, and use the inhibitor as only one manipulation in a larger study would dramatically increase the overall impact of the study. As a simple cachectic prevention design, most measured changes in variables found in muscle and fat are related to the fact that the inhibitor lowered circulating IL-6, maintained food intake, and muscle mass loss was mitigated. A weight stable muscle that does not activate cellular wasting signaling is not novel.

This mechanistic deficit could be offset by many potential additional experiments, for example giving the inhibitor after weight loss as been initiated, examining pre-cachectic tumor bearing mice. A standard in the field is that the mechanistic examination of muscle wasting during cachexia includes treatments that do not affect the underlying disease, and experiments that can determine direct versus systemic effects on muscle signaling. The systemic the inhibitor centered study fails on both counts.

We performed additional *in vivo* experiments in which we started (+)-JQ1 treatment after C26-tumor bearing mice started to lose weight. The obtained results support our initial findings and the manuscript conclusions because the survival curve is not significantly different when animals are treated since d1 or after they lost 10% of their weight. Please see our detailed response below in reference to the Specific part of the Reviewer comments, which in manuscript is contained in the Results section (pages 5-6).

Please see below, our responses to the specific comments.

Specific

Introduction: A stronger premise in the introduction is needed. While it is mentioned in the manuscript several times that there are no approved treatments for cancer cachexia, a proposed reason for this fact is not presented, cachexia has been widely investigated for 25 years. Furthermore, how the current inhibitor study overcomes this gap is not

clearly stated. IL-6, STAT3, E3 ligases, autophagy and even AMPK have been widely investigated as potential therapeutic targets.

Thank you for pointing this to us, this is a very relevant comment. We have introduced a paragraph underscoring that the multifaceted pathophysiology of cancer cachexia requires a multimodal therapeutic approach. In this scenario several targets such as IL6, ubiquitin proteasome system and autophagy have been investigated for therapeutic intervention in the past but with limited success. We suggest that manipulation of epigenetic modulators targeting both inflammation and skeletal muscle catabolic pathways may be more effective in tempering multiple targets in different tissues.

Please see page 3: “In humans, therapeutic intervention with a single agent targeting exclusively systemic inflammation, low food intake, catabolic pathways regulating muscle mass has proven ineffective to counteract cachexia^{6,7}. A combinatorial approach is advocated for successful management of cachexia, in order to concurrently address different facets of the syndrome, such as inflammation and muscle atrophy⁸. IL6, the ubiquitin proteasome system, and autophagy have been extensively investigated for potential therapeutic intervention, with limited success^{9-11,12}. Thus, the precise elucidation of molecular pathways underlying cancer cachexia is crucial for the identification of novel key targets, which regulate multiple pathophysiological aspects of the disease and that become particularly appealing for therapeutic intervention. Transcription factors and epigenetic regulators are of particular interest in this scenario, as they hold the potential to reprogram multiple transcription programs in different tissues and orchestrate coordinated transcription in different districts^{6,7,12}.”

-Introduction: First sentence needs a reference.

We have provided a reference for first sentence. Please see page 3: “Cancer cachexia is a multifactorial metabolic syndrome characterized by systemic inflammation and muscle and adipose tissue wasting, which lead to weight loss despite adequate nutritional support¹. ”

Results: Why was the in vivo study stopped at 12 days when the animals survived another 5-7 days?

In our survival experiment, vehicle-treated C26-tumor bearing animals started to die at d15 (1 mouse) and at d16 we lost 3 additional animals, see Fig. 2C and Fig. S2A, in the revised manuscript.

At day 12, vehicle treated C26-tumor bearing animals lose an average of 22% of body weight, which corresponds to severe cachectic condition. In fact, cancer cachexia evaluation in mouse experimental models is usually divided in three stages: initial low ($\leq 5\%$), intermediate (6–19%), or severe ($\geq 20\%$) body weight loss¹³. We agree that in the literature different periods of time are used to reach the severe stage of cachexia, and this can be due to different factors: age of mice at tumor inoculation, mouse strains, C26 sub-clones selection. Nonetheless, the 12 days window is not unusual and was employed in several other studies^{9,14,15}. Additionally, to comply with the ethic requirements of our animal protocol we have to sacrifice animals at 20% weight loss. We have added this information, in the Methods section, on page 18.

What was the advantage examining 48-72 window of cachexia?

In the C26 experimental model mice develop cachexia rapidly. For ethical reasons, restriction in our IACUC protocol would not allow to extend our examination window, because of the severe weight loss (above 20%).

We have included body weights related to the survival experiments, in Fig. S2A. These data show that weight stabilization is maintained for several days, suggesting that cachexia is hampered and delayed (grey line versus black line).

Additionally, we were able to observe partial cachexia reversal when (+)-JQ1 treatment was started at 10% weight loss (blue line versus black line).

Overall, these observations suggest that the alleviation of cachexia is maintained for several days, upon (+)-JQ1 treatment.

Why were tumor bearing non-cachectic mice not included in the analysis 10 days?

Tumor bearing non-cachectic mice were not included in the analysis at day 10, because the genes expression program for catabolic pathways is not active yet.

Based on our BRD4 ChIP-Seq data in control animals, we expect that at earlier stages catabolic pathways are only slightly activated and that BRD4 recruitment is very low (Fig. 4C-E). We expect BRD4 to engage atrogenes regulatory regions at later stages of cachexia, when catabolic pathways are actively transcribed and BRD4 potentiates their transcription. Therefore, analysis at d10 would be not very informative, because of the likely limited BRD4 engagement at genes promoting catabolic pathways.

Why did the study not examine the benefits of the inhibitor administrated after the initiation of weight loss on survival, which is more clinically relevant to a therapeutic target?

As we stated above, we agreed that the issue raised by the reviewer is highly relevant from a clinical perspective and we performed the new experiment and included data in Fig. 2C and S2A. The obtained results are very supportive for the conclusions of the manuscript since the survival significantly increases when the inhibitor administration starts at 10% of weight loss (moderate cachexia), and survival is comparable to the one obtained when treatment started at day 1. These data suggest that JQ1 administration can partially revert catabolic pathways activation and lead to a weight stabilization even at moderate cachexia condition.

Please see pages 5-6: “To provide a more therapeutically relevant preclinical model, we started (+)-JQ1 administration at a moderate cachexia stage, when C26-tumor bearing mice displayed an average 10% of weight loss. Surprisingly, (+)-JQ1 treatment was able to significantly prolong survival and partially reverse cachexia, shifting median survival from 16 to 28 days (Fig. 2C).

When JQ1 was administered the day after tumor cells inoculation, cachexia was delayed and C26-tumor bearing mice started to lose weight at day 24 (Fig. S2A, grey line). Furthermore, when treatment was started during moderate cachexia, C26-tumor bearing mice partially reversed weight loss few days after (+)-JQ1 treatment started, and their weight was stabilized till day 23, further delaying cachexia (Fig.S2A, blue line).”

Results: Based on the survival data can the reader assume that the inhibitor only delayed the onset of cachexia? Please provide a justification for your answer, as this would be of interest to those trying to interpret the findings.

JQ1 administration prolonged survival, both when treatment started at day 1 and at moderate cachexia. Treating the animals at day 1 delays the onset of cachexia, since weight loss starts at day 16, 7 days later than in vehicle treated mice, and even at death (+)-JQ1 treated animals weigh more than vehicle treated C26 tumor bearing mice. While weight loss is stabilized, tumors keep growing, they ulcerate and the overall animal conditions slowly worsen, leading to death.

Furthermore, when JQ1 administration was started at 10% weight loss we had a partial reversal of cancer cachexia, since (+)-JQ1-treated animals start to gain weight back 3 days after treatment began, and their weight is stabilized for approximately 10-11 days. In this latter experiment therefore, we do not observe a delay but a partial recovery and reversal of cachexia. Body weights of animals used in Fig. 2C are reported in Fig. S2A. This justification is now reflected in the paragraph of the revised manuscript see pages 5-6. "To provide a more therapeutically relevant preclinical model, we started (+)-JQ1 administration at a moderate cachexia stage, when C26-tumor bearing mice displayed an average 10% of weight loss. Surprisingly, (+)-JQ1 treatment was able to significantly prolong survival and partially reverse cachexia, shifting median survival from 16 to 28 days (Fig. 2C).

When JQ1 was administered the day after tumor cells inoculation, cachexia was delayed and C26-tumor bearing mice started to lose weight at day 24 (Fig. S2A, grey line). Furthermore, when treatment was started during moderate cachexia, C26-tumor bearing mice partially reversed weight loss few days after (+)-JQ1 treatment started, and their weight was stabilized till d23, further delaying cachexia (Fig.S2A, blue line)."

Results: Is it surprising that catabolic pathways are not activated in a non-wasting muscle? More exciting would be catabolic pathways activated that no longer lead to wasting, do the authors have any information in this area?

This point was clarified by the treatment after weight loss initiated.

BRD4 and BRD2 are chromatin factors that modulate transcriptional regulation of their targets; consequently BRD4/BRD2 blockade lead to decreased expression of their targets. At several loci this regulation occurs at super-enhancer regions, which represent chromatin regions particularly dense in transcription factors and chromatin factors, and which sustain the upregulation of downstream genes. Thus, when JQ1 is administered since day 1, catabolic pathways are not activated in JQ1 treated mice. Nevertheless, in animals treated after weight loss has started we expect that catabolic pathways were transcriptionally active and that the BET inhibitor was able to block transcriptional activation of catabolic genes and therefore muscle degradation.

Can the authors provide another AMPK downstream phosphorylation target other than FOXO, there are many and this would strengthen the interpretation?

Two other AMPK-specific phosphorylation targets are included: Beclin1 (Fig. S3) and Ulk1 (Fig. 3J) and mentioned on page 7.

Results. epididymal adipose section: If the author's feel this is an important point for the manuscript some of the adipose wasting data should be moved from the supplement to the actual paper, or discussion of this point minimized

We agreed with Reviewer and we moved the data related to adipose tissue to the paper, Figure 8.

Figure 8 data and text. There needs to be some descriptor data of the patients and weight loss. Please add in text or provide in supplement to help with interpretation.

We added a descriptor of patients' data in supplemental information, Suppl. Table 2.

Methods: Provide the N analyzed for each figure or assay in the legend.

We included n in Figures 5G and H (n=3) and Fig S6 D and E (n=3).

Statistical analysis: related to rigor, please clarify where unpaired t-tests and ANOVA analysis were used.

ANOVA followed by post-hoc Tukey analysis was used when experimental groups were equal or more than 3. Unpaired t-test was used when we compared 2 experimental groups. This information was added in the Statistical Analysis description and in each figure legend.

Please see page 21:” When we compared 2 experimental groups we used unpaired Student's t test and when we compared 3 or more experimental groups we used one-way analysis of variance (ANOVA) followed by the Tukey's post hoc test.”

Figure 5G & 5H. Long-term compound C administration has often been reported to be toxic to myotubes. Please include the Compound C only controls.

In fig. 5G and 5H, we reported myotubes stimulated with Compound C alone, as a control.

Figure 7D. Please clarify if insulin levels were controlled for. It is also of interest why defined cachectic media from tumor cells was not used in place of serum starvation. Please clarify.

Insulin levels were controlled for, since insulin was not added in differentiation medium. The reason is that we did not want to obtain confounding results on FoxO3 by activating the AKT pathway. Please see Methods description on page 16.

We repeated experiments previously performed with starvation using C26- conditioned medium. We agree that while the starvation model allow us to broaden our findings to other well-established atrophy models, the latter system better mimics C26-induced cachexia conditions. Please see Fig. S11E with description included in page 12.

Page 12, first sentence: an incomplete sentence

Thank you, we corrected the incomplete citation.

Please see page 13: “Cancer patients were designated as cachectic according to the definition by Fearon et al., (2011)¹⁶.”

Reviewer #2:BRD4 inhibitors and epigenetic

(Remarks to the Author):The manuscript by Caretti and co-workers describes the positive effects of the pan-BET inhibitor JQ1 in a C-26 colon carcinoma model of cancer cachexia, preventing weight loss of tumor-bearing mice and reducing muscle and adipose tissue wasting. The effect is accompanied by increased survival rate of the animals in this model. The BET bromodomain BRD4 is at least in part responsible for this effect by regulating the cytokine IL-6, a well-known downstream target of BRD4, as well as through modulation of the AMPK pathway and prevention of phosphor-AMPK nuclear accumulation. Transcriptional regulation of FOXO3, a key transcription factor of catabolic pathways, was also affected by (+)-JQ1 treatment. Moreover treatment with (+)-JQ1 prevented UPS and autophagy induction muscles of C26-tumor bearing mice. Based on their recent finding that BRD4 plays a role in controlling myotube size and that (+)-JQ1 was able to prevent atrophy of myotubes in a dexamethasone induced model the authors investigated if BRD4 is involved in transcriptional regulation of cachexia induced catabolic genes. ChipSeq experiments confirmed the association of BRD4 with regulatory sites in several catabolic genes. Treatment of mice with (+)- JQ1, but not the negative control compound (-)-JQ1 prevented the activation of catabolic genes associated with skeletal muscle atrophy and decreased IL6 systemic levels. They also show that in biopsies from cachexic patients these pathways are relevant. The finding that BRD4 plays a role in cachexia is new and the author provide a comprehensive analysis of the pathway involved and its regulation by BET bromodomains. Given the fact that BET inhibitors are already in several clinical trials in oncology this study is highly relevant and timely. I recommend the manuscript for publication pending some additional amendments:

Reviewer 2: We greatly appreciate the positive assessment and recommendation expressed by the reviewer.

Abstract: (+)-JQ1 protects tumor-bearing mice from body weight loss, muscle and adipose tissue wasting. Remarkably (+)-JQ1 administration dramatically prolongs survival without directly affecting tumor growth. – This statement is slightly misleading as it is phrased, implicating that the lack of affect in tumor proliferation is a general effect. It should be rephrased in a way to make it clear that in a single cell line (+)-JQ1 did not affect tumor growth and that this cell line was used for further studies of cachexia.

We agree that the statement was misleading and corrected the sentence in the abstract.

Please see page 2: “Remarkably, in C26-tumor bearing mice (+)-JQ1 administration dramatically prolongs survival, without directly affecting tumor growth. By ChIP-seq analyses, we unveil that the BET proteins directly promote the muscle atrophy program during cachexia.”

- *What is the method used for tumor cell antiproliferative effect in Fig 1? Which cancer is associated with the different cell lines used in Fig 1. Which method was used to measure proliferation?*

100000cells were treated with JQ1 or vehicle for 72hrs. After 72hrs, cells were colored with trypan blue and cells were counted, to evaluate cell viability. Cell viability was

represented showing the percentage of viable cells (control cell=100% viability). We corrected “number of cells” with cell viability in the graphs (Fig. 1A) and added the information above in the methods section.

See page 16: “After 72hrs, cells were colored with trypan blue and viable cells were counted, to evaluate cell viability. Cell viability was represented showing the percentage of viable cells (control cell=100% viability).”

- *Fig 2 treadmill test: why was the test not performed using (-)-JQ1 tested? Fig 2D: is there statistical significance between vehicle-treated animals and animals treated with (-)-JQ1. If so what is the rationale for this? Likewise in Figure 3A*

The initial characterization was performed with two doses of (+)- JQ1 and (-)-JQ1 and vehicle. We observed that morphological, histological and molecular biology data were fully comparable in (-)-JQ1 and vehicle treated animals. Therefore, to comply with the 3R rule and use the necessary number of animals to reach statistical significance, we restricted our studies to vehicle and (+)-JQ1-treated animals in the treadmill test.

Below we report a snapshot of statistical calculation performed by Prism6 (GraphPad) for Figures 2D and 3A, showing that there is no statistical difference between vehicle and (-)-JQ1-treated animals.

Fig. 2D Epididymal adipose tissue

Table Analyzed	Data 1		
One-way analysis of variance			
P value	< 0,0001		
P value summary	***		
Are means signif. different? (P < 0.05)	Yes		
Number of groups	6		
F	22.54		
R square	0.6843		
Bartlett's test for equal variances			
Bartlett's statistic (corrected)	11.94		
P value	0.0356		
P value summary	*		
Do the variances differ signif. (P < 0.05)	Yes		
ANOVA Table			
	SS	df	MS
Treatment (between columns)	419488	5	83898
Residual (within columns)	193570	52	3723
Total	613058	57	
Tukey's Multiple Comparison Test			
	Mean Diff,	q	Significant? P < 0,05?
Control vs C26 + veh	187.3	9.708	Yes
Control vs C26 + (-)-JQ1 20	226.3	11.06	Yes
Control vs C26 + (-)-JQ1 50	210.1	10.89	Yes
Control vs C26 + (+)-JQ1 20	59.44	3.081	No
Control vs C26 + (+)-JQ1 50	72.68	3.767	No
C26 + veh vs C26 + (-)-JQ1 20	38.96	1.904	No
C26 + veh vs C26 + (-)-JQ1 50	22.79	1.181	No
C26 + veh vs C26 + (+)-JQ1 20	-127.9	6.627	Yes
C26 + veh vs C26 + (+)-JQ1 50	-114.6	5.941	Yes
C26 + (-)-JQ1 20 vs C26 + (-)-JQ1 50	-16.17	0.7902	No
C26 + (-)-JQ1 20 vs C26 + (+)-JQ1 20	-166.8	8.152	Yes
C26 + (-)-JQ1 20 vs C26 + (+)-JQ1 50	-153.6	7.505	Yes
C26 + (-)-JQ1 50 vs C26 + (+)-JQ1 20	-150.7	7.808	Yes
C26 + (-)-JQ1 50 vs C26 + (+)-JQ1 50	-137.4	7.122	Yes
C26 + (+)-JQ1 20 vs C26 + (+)-JQ1 50	13.24	0.6862	No

Fig. 3A fast Myosin Heavy Chain

Table Analyzed		Data 1				
One-way analysis of variance						
P value	< 0,0001					
P value summary	***					
Are means signif. different? (P < 0.05)	Yes					
Number of groups	6					
F	22.91					
R square	0.8642					
ANOVA Table		SS	df	MS		
Treatment (between columns)		2.029	5		0.4059	
Residual (within columns)		0.3189	18		0.01772	
Total		2.348	23			
Tukey's Multiple Comparison Test		Mean Diff,	q	Significant? P < 0,05?	Summary	95% CI of diff
Control vs C26 + veh		0.5725	8.602	Yes	***	0,2733 to 0,8717
Control vs C26 + (-)-JQ1 20		0.6725	10.1	Yes	***	0,3733 to 0,9717
Control vs C26 + (-)-JQ1 50		0.4925	7.4	Yes	***	0,1933 to 0,7917
Control vs C26 + (+)-JQ1 20		0.065	0.9766	No	ns	-0,2342 to 0,3642
Control vs C26 + (+)-JQ1 50		-0.035	0.5259	No	ns	-0,3342 to 0,2642
C26 + veh vs C26 + (-)-JQ1 20		0.1	1.503	No	ns	-0,1992 to 0,3992
C26 + veh vs C26 + (-)-JQ1 50		-0.08	1.202	No	ns	-0,3792 to 0,2192
C26 + veh vs C26 + (+)-JQ1 20		-0.5075	7.625	Yes	***	-0,8067 to -0,2083
C26 + veh vs C26 + (+)-JQ1 50		-0.6075	9.128	Yes	***	-0,9067 to -0,3083
C26 + (-)-JQ1 20 vs C26 + (-)-JQ1 50		-0.18	2.705	No	ns	-0,4792 to 0,1192
C26 + (-)-JQ1 20 vs C26 + (+)-JQ1 20		-0.6075	9.128	Yes	***	-0,9067 to -0,3083
C26 + (-)-JQ1 20 vs C26 + (+)-JQ1 50		-0.7075	10.63	Yes	***	-1,007 to -0,4083
C26 + (-)-JQ1 50 vs C26 + (+)-JQ1 20		-0.4275	6.423	Yes	**	-0,7267 to -0,1283
C26 + (-)-JQ1 50 vs C26 + (+)-JQ1 50		-0.5275	7.926	Yes	***	-0,8267 to -0,2283
C26 + (+)-JQ1 20 vs C26 + (+)-JQ1 50		-0.1	1.503	No	ns	-0,3992 to 0,1992

- *P.8 2nd paragraph; correct spelling of: 'J(+)-Q1' restrains cancer cachexia through the reduction of IL6...*

We corrected the misspelling.

- *BRD2, another Bet family protein is potently inhibited by (+)-JQ1 as well. BRD2 has been linked to metabolism with BRD2 loss in mice causing obesity. The author should show if BRD2 is involved in cachexia.*

JQ1 binds to BRD4 with a Kd of about 50nM and significantly higher Kd (128nM) to BRD2¹⁷. It is possible that JQ1 targets both BRD4 and BRD2 in our experiments.

To address this point, we performed BRD2 ChIP experiments in skeletal muscle and in C26 tumors, and investigated BRD2 engagement on genes promoting muscle catabolism and on the IL6 and PTHrP promoters, respectively. Data are reported in Supplemental material: please see Fig. S6A and S8A, and pages 8 and 9 for results presentation.

- 1 Urbanucci, A. *et al.* Androgen Receptor Deregulation Drives Bromodomain-Mediated Chromatin Alterations in Prostate Cancer. *Cell Rep* **19**, 2045-2059, doi:10.1016/j.celrep.2017.05.049 (2017).
- 2 Fernandez-Alonso, R. *et al.* Brd4-Brd2 isoform switching coordinates pluripotent exit and Smad2-dependent lineage specification. *EMBO Rep* **18**, 1108-1122, doi:10.15252/embr.201643534 (2017).

- 3 Cheung, K. L. *et al.* Distinct Roles of Brd2 and Brd4 in Potentiating the
Transcriptional Program for Th17 Cell Differentiation. *Mol Cell* **65**, 1068-
1080.e1065, doi:10.1016/j.molcel.2016.12.022 (2017).
- 4 Fujimoto-Ouchi, K., Onuma, E., Shirane, M., Mori, K. & Tanaka, Y.
Capecitabine improves cancer cachexia and normalizes IL-6 and PTHrP levels in
mouse cancer cachexia models. *Cancer Chemother Pharmacol* **59**, 807-815,
doi:10.1007/s00280-006-0338-y (2007).
- 5 Kir, S. *et al.* PTH/PTHrP Receptor Mediates Cachexia in Models of Kidney
Failure and Cancer. *Cell Metab* **23**, 315-323, doi:10.1016/j.cmet.2015.11.003
(2016).
- 6 Aversa, Z., Costelli, P. & Muscaritoli, M. Cancer-induced muscle wasting: latest
findings in prevention and treatment. *Ther Adv Med Oncol* **9**, 369-382,
doi:10.1177/1758834017698643 (2017).
- 7 Fearon, K. C., Glass, D. J. & Guttridge, D. C. Cancer cachexia: mediators,
signaling, and metabolic pathways. *Cell Metab* **16**, 153-166, doi:S1550-
4131(12)00248-3 [pii]
10.1016/j.cmet.2012.06.011 (2012).
- 8 Maddocks, M. *et al.* Practical multimodal care for cancer cachexia. *Curr Opin
Support Palliat Care* **10**, 298-305, doi:10.1097/SPC.0000000000000241 (2016).
- 9 Penna, F. *et al.* Effect of the specific proteasome inhibitor bortezomib on cancer-
related muscle wasting. *J Cachexia Sarcopenia Muscle* **7**, 345-354,
doi:10.1002/jcsm.12050 (2016).
- 10 Narsale, A. A. & Carson, J. A. Role of interleukin-6 in cachexia: therapeutic
implications. *Curr Opin Support Palliat Care* **8**, 321-327,
doi:10.1097/SPC.0000000000000091 (2014).
- 11 Au, E. D., Desai, A. P., Koniaris, L. G. & Zimmers, T. A. The MEK-Inhibitor
Selumetinib Attenuates Tumor Growth and Reduces IL-6 Expression but Does
Not Protect against Muscle Wasting in Lewis Lung Cancer Cachexia. *Front
Physiol* **7**, 682, doi:10.3389/fphys.2016.00682 (2016).
- 12 Sakuma, K., Aoi, W. & Yamaguchi, A. Molecular mechanism of sarcopenia and
cachexia: recent research advances. *Pflugers Arch* **469**, 573-591,
doi:10.1007/s00424-016-1933-3 (2017).
- 13 White, J. P. *et al.* The regulation of skeletal muscle protein turnover during the
progression of cancer cachexia in the Apc(Min/+) mouse. *PLoS One* **6**, e24650,
doi:10.1371/journal.pone.0024650 (2011).
- 14 Bonetto, A. *et al.* STAT3 activation in skeletal muscle links muscle wasting and
the acute phase response in cancer cachexia. *PLoS One* **6**, e22538,
doi:10.1371/journal.pone.0022538 (2011).
- 15 Sciorati, C. *et al.* Necdin is expressed in cachectic skeletal muscle to protect fibers
from tumor-induced wasting. *J Cell Sci* **122**, 1119-1125, doi:10.1242/jcs.041665
(2009).
- 16 Fearon, K. *et al.* Definition and classification of cancer cachexia: an international
consensus. *Lancet Oncol* **12**, 489-495, doi:10.1016/S1470-2045(10)70218-7
(2011).
- 17 Filippakopoulos, P. *et al.* Selective inhibition of BET bromodomains. *Nature* **468**,
1067-1073, doi:10.1038/nature09504 (2010).

Reviewer #1 (Remarks to the Author):

This is a revised manuscript linking BRD4 to IL-6 / AMPK / FOXO3 signaling that regulates catabolic gene regulation during cancer cachexia. The study employs in vitro and in vivo analysis using preclinical models of cancer cachexia. Although there were several identified strengths in the original version of the manuscript, weaknesses were identified that affected the predicted impact of the study. The authors have been extremely responsive to the previously identified weaknesses and have added additional experiments, clarification in the text, and thoughtful rebuttals. These additional experiments and responses serve to strengthen the expected impact of the study, and also the mechanistic insight provided by the data presented.

The authors have added new ChIP assay experiments that improve the study's mechanistic insight. The text has been revised to provide a stronger premise and discuss potential systemic and local effects. The added in vivo experiment, starting treatment after weight loss, significantly increases the clinical relevance of the findings.

Comment

It does not appear the additional in vivo treatment was added to the methods section of the manuscript. Please add the method detail to the animal section related to the experiment administering JQ1 after the initiation of weight loss.

Reviewer #2 (Remarks to the Author):

The authors have adequately addressed all points raised and I recommend the article for publication.

Rebuttal to Reviewers' comments.

We are happy to learn that both reviewers found our revised manuscript significantly improved and that we were able to address their concerns.

In response to Reviewer#1 Comment, we have included in the manuscript information regarding the in vivo experiment in which JQ1 was administered when weight loss already started, in Methods section, page 19.

“In survival studies, when treatment was started at moderate cachexia, (+)-JQ1 (20 mg/kg body weight) or vehicle administration began when C26-tumor bearing mice average weight loss was 10%. Animal weight and food intake were recorded daily.”

Reviewer #1 (Remarks to the Author):

This is a revised manuscript linking BRD4 to IL-6 / AMPK / FOXO3 signaling that regulates catabolic gene regulation during cancer cachexia. The study employs in vitro and in vivo analysis using preclinical models of cancer cachexia. Although there were several identified strengths in the original version of the manuscript, weaknesses were identified that affected the predicted impact of the study. The authors have been extremely responsive to the previously identified weaknesses and have added additional experiments, clarification in the text, and thoughtful rebuttals. These additional experiments and responses serve to strengthen the expected impact of the study, and also the mechanistic insight provides by the data presented.

The authors have added new ChIP assay experiments that improve the study's mechanistic insight. The text has been revised to provide a stronger premise and discuss potential systemic and local affects. The added in vivo experiment, starting treatment after weight loss, significantly increases the clinical relevance of the findings.

Comment

It does not appear the additional in vivo treatment was added to the methods section of the manuscript. Please add the method detail to the animal section related to the experiment administering JQ1 after the initiation of weight loss.

Reviewer #2 (Remarks to the Author):

The authors have adequately addressed all points raised and I recommend the article for publication.